# System Prompt Optimization with Meta-Learning

**Yumin Choi**[1*]   **Jinheon Baek**[1*]   **Sung Ju Hwang**[1,2]
KAIST[1],  DeepAuto.ai[2]
{yuminchoi, jinheon.baek, sungju.hwang}@kaist.ac.kr

## Abstract

Large Language Models (LLMs) have shown remarkable capabilities, with optimizing their input prompts playing a pivotal role in maximizing their performance. However, while LLM prompts consist of both the task-agnostic system prompts and task-specific user prompts, existing work on prompt optimization has focused on user prompts specific to individual queries or tasks, and largely overlooked the system prompt that is, once optimized, applicable across different tasks and domains. Motivated by this, we introduce the novel problem of bilevel system prompt optimization, whose objective is to design system prompts that are robust to diverse user prompts and transferable to unseen tasks. To tackle this problem, we then propose a meta-learning framework, which meta-learns the system prompt by optimizing it over various user prompts across multiple datasets, while simultaneously updating the user prompts in an iterative manner to ensure synergy between them. We conduct experiments on 14 unseen datasets spanning 5 different domains, on which we show that our approach produces system prompts that generalize effectively to diverse user prompts. Also, our findings reveal that the optimized system prompt enables rapid adaptation even to unseen tasks, requiring fewer optimization steps for test-time user prompts while achieving improved performance.

## 1   Introduction

Large Language Models (LLMs) have demonstrated remarkable capabilities across a wide range of tasks and domains [1, 7, 23]. To effectively harness LLMs in diverse application scenarios, prompts play a pivotal role in guiding their behavior and ensuring their outputs align with user goals, which comprise two components: system prompts and user prompts [25]. Specifically, system prompts are task-agnostic instructions that define the foundational behavior and constraints of the LLM (which are also designed to be applicable to multiple tasks and domains); whereas, user prompts are task-specific inputs designed to elicit responses tailored to solving particular queries or tasks.

Alongside the advancement of LLMs, as their performance is highly sensitive to the prompts provided, there has been a surge of interest in designing effective prompts. Traditionally, manual prompt crafting has been the dominant approach, which has led to the discovery of several prompts, such as Chain-of-Thought, which is known for enhancing the reasoning capabilities of LLMs [39]. However, this manual design process is labor-intensive and limited in scalability. Therefore, to overcome these limitations, the field of automatic prompt optimization has emerged, which aims to automatically improve prompts by utilizing LLMs directly or further integrating them with algorithms to explore and generate more effective prompt variations [12, 14, 41, 45]. Specifically, notable methods include textual gradients [27], which produces gradients (that criticize prompts) in text based on model prediction results and formulates new prompts iteratively, and the approach with Monte Carlo Tree Search (MCTS) [37], which explores and evaluates various prompt configurations through tree search.

---

*Equal contribution; Code is available at https://github.com/Dozi01/MetaSPO.

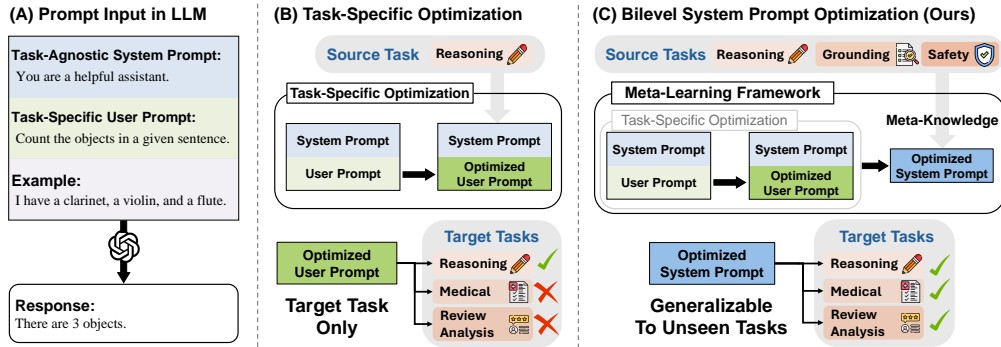

Figure 1: **Concept Figure.** (A) The input prompt provided to LLMs typically consists of a task-agnostic system prompt, a task-specific user prompt, and a target example to handle. (B) Conventional Task-Specific Optimization focuses on optimizing user prompts for a single task but shows limited generalization to other tasks. (C) The goal of Bilevel System Prompt Optimization (Ours) is to enable the optimized system prompt to generalize effectively to unseen target tasks, for which we utilize a meta-learning framework to derive meta-knowledge from multiple source tasks.

However, despite various studies on prompt optimization, they have mainly focused on user prompts for solving a specific task, while overlooking the system prompts that could significantly influence response generation and have far-reaching effects across diverse tasks and domains. Notably, there are a couple of benefits in optimizing system prompts. First, system prompts serve as the foundational instructions that are invariant, meaning that a single well-optimized system prompt can generalize across multiple tasks and domains. Second, the optimized system prompts can establish a robust behavioral framework, enabling LLMs to adapt more robustly to unseen user prompts and domains, while having the potential to create a synergistic relationship with user prompts.

To address this gap, we introduce the novel problem of bilevel system prompt optimization (Figure 1), which aims to design system prompts that can be effectively coupled with diverse user prompts and generalizable across a wide range of tasks, including those not seen during optimization. However, in contrast to conventional approaches that focus on optimizing only the user prompts, the proposed problem introduces unique challenges as it requires optimizing two objectives (system and user prompts) simultaneously. To handle this, we then frame it as a bilevel optimization framework, as its hierarchical structure allows us to decouple the optimization processes (for system and user prompts) while capturing their dependency. Intuitively, within the bilevel optimization, the system prompt (optimized to generalize for diverse tasks) forms the higher-level optimization objective, while the user prompts (optimized to maximize task-specific performance) form the lower-level objective.

Building on this bilevel formulation, we then propose to tackle the aforementioned problem of system prompt optimization through a meta-learning framework, which is particularly well-suited as it learns to generalize from a distribution of tasks (rather than individual queries and tasks) and subsequently enables the robust and rapid adaptation to various user prompts and tasks. Specifically, in our problem setup, the proposed meta-learning framework meta-trains the system prompt to be optimized over a diverse range of user prompts and tasks (via the higher-level optimization objective), equipping it with the ability to generalize even to unseen instances. Furthermore, by iteratively updating the user prompts through the lower-level optimization objective within the meta-learning loop, our approach ensures that the system prompt is optimized in synergy with diverse user prompts. Additionally, this meta-learning design choice offers a couple of advantages. First, compared to approaches that are not on meta-learning [43], the proposed framework is designed to be superior to them in handling various unseen prompts and tasks, which are prevalent in real-world scenarios. Also, the framework is highly versatile since it allows for the use of any prompt optimization techniques to operationalize it. We refer to our overall method as Meta-level System Prompt Optimizer (in short, **MetaSPO**).

We extensively validate our MetaSPO framework on 14 unseen tasks across 5 diverse domains under two different real-world scenarios: 1) unseen generalization, where the optimized system prompt is directly leveraged to test time queries without any further optimization of user prompts; 2) test-time adaptation, where the user prompts (specific to target tasks) are further optimized with few examples from them (while the optimized system prompt remains fixed). From this, we then observe that, in

the unseen generalization scenario, the optimized system prompt significantly outperforms baseline methods, demonstrating its strong generalization capabilities across diverse, unseen tasks and user prompts. Additionally, in the test-time adaptation scenario, the optimized system prompt facilitates more efficient user prompt optimization, leading to faster convergence with superior performance.

## 2    Related Work

**System Prompts**    System prompts, introduced in ChatGPT [25], have become an integral part of modern LLMs, playing a crucial role in defining foundational behavior [2, 11, 40]. As their adoption grows, many studies on system prompts have begun to uncover their potential. To mention a few, Zheng et al. [44] demonstrates that incorporating a persona into system prompts can enhance LLM performance on certain role-playing scenarios. Additionally, some recent studies propose training techniques to enhance model alignment with various system prompts [19, 11]. In particular, Wallace et al. [35] suggests training the model to follow instructions with the highest importance, written in system prompts, with specific data generation strategies for it. However, unlike the aforementioned studies that aim to align and test models to pre-defined system prompts (which are handcrafted), our work focuses on automatically optimizing system prompts.

**Prompt Optimization**    As the performance of LLMs is highly sensitive to the quality and structure of their prompts, the field of prompt optimization has received much attention. Early studies rely on gradient-based methods to adjust a small number of trainable parameters and inject them into LLMs as an embedding-level soft prompt; however, they are computationally expensive and unsuitable for use with closed-source models [22, 20, 38]. To address this, gradient-free methods have emerged, whose core idea is to generate candidate prompts with LLMs and evaluate them iteratively to select the most effective one, as in APE [45] and OPRO [41]. Also, there exist methods that additionally perform problem analysis for the current prompt before crafting the optimized prompts [27, 8, 37]. Furthermore, some of the other works leverage the idea of evolutionary algorithms (like crossover and mutation) in optimizing prompts with LLMs [14, 12, 9]. Despite these advancements, existing studies have centered on optimizing user prompts specific to certain tasks, with limited exploration of system prompts. Also, although very recent studies have started exploring the system prompt, either their focus is restricted to only the safety-related tasks [46] or lacks consideration of interactions with diverse user prompts [43], despite the fact that the optimized system prompt should be generalizable over diverse tasks with various user prompts. To this end, we approach to address this gap through the novel formulation of bilevel system prompt optimization, and tackle it with meta-learning.

**Meta-Learning**    Meta-learning, or the concept of learning to learn, aims to acquire generalizable knowledge across a distribution of tasks and enable models to adapt to new tasks with no or minimal training, unlike conventional approaches that optimize for a single task or dataset. To be specific, the approach called Model-Agnostic Meta-Learning (MAML) learns a shared initialization that, when used for fine-tuning, enables rapid adaptation across tasks [13]. Also, Matching Networks [34] and Prototypical Networks [32] first represent task distributions over the embedding space by mapping and learning samples over it, and then use this learned embedding for adaptation. Recently, meta-learning has also been adopted in the domain of prompt optimization, but prior works have focused only on gradient-based methods (fine-tuning trainable parameters), orthogonal to our focus on gradient-free approaches [38, 28, 16]. Also, meta-learning has not been explored for system prompt optimization.

## 3    Methodology

We begin with preliminaries, providing a formal explanation of Large Language Models (LLMs) and conventional prompt optimization techniques. We then introduce the problem of bilevel system prompt optimization and propose a meta-learning-based approach to tackle it.

### 3.1    Preliminary

**Large Language Models**    Formally, LLMs take the input $x$, which typically consists of a system prompt $s$, a user prompt $u$, and a specific query to respond $q$, then generate the output $y$, formalized as follows: $y = \text{LLM}(x)$, where $x = [s, u, q]$ and each element (such as $s$, $u$, $q$, $x$, and $y$) is represented as a sequence of tokens, e.g., $s = [s_1, \ldots, s_i]$. In this formulation, the system prompt

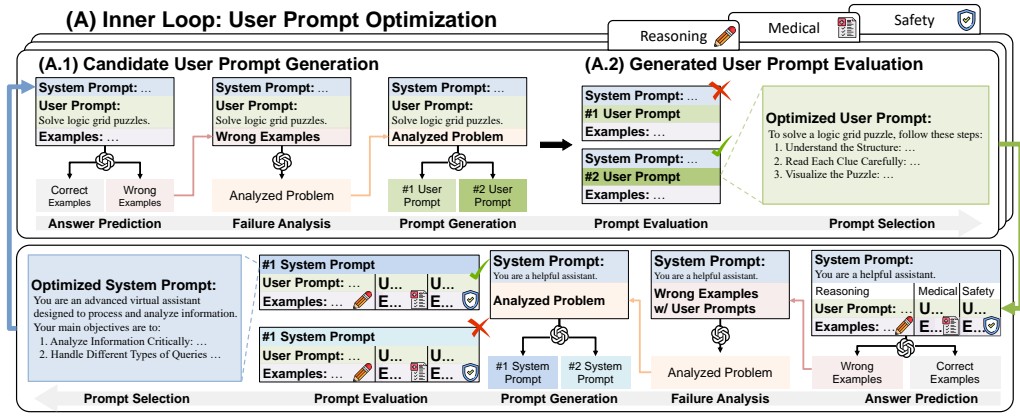

Figure 2: **Overview of MetaSPO**, which consists of the inner loop for user prompt optimization and the outer loop for system prompt optimization, operationalized through the meta-learning framework. (A) Inner Loop generates candidate user prompts by analyzing incorrectly predicted examples and then evaluates them with the system prompt to select refined prompts for individual tasks. (B) Outer Loop generates candidate system prompts by analyzing incorrect examples from all source tasks, and then evaluates them across various user prompts and tasks to ensure generalizability.

$s$ defines high-level behavioral guidelines that are designed to be task-agnostic, the user prompt $u$ specifies the fine-grained task or action to be performed, and the query $q$ contains the specific input or instance that requires a response, all of which allow the LLM to produce the desired output $y$.

**Prompt Optimization**   Given the sensitivity of LLMs to the prompts they receive, prompt optimization emerges as an effective solution, whose objective is to automatically discover prompts that maximize task performance through iterative refinement processes, guided by performance feedback from available examples. Formally, let $T$ denote a task, which represents a dataset or distribution of input-output pairs $(q, a)$, where $q$ is the query and $a$ is the ground truth answer. Then, the goal of prompt optimization is to identify the optimal prompt $u^*$ (starting from the initial prompt $u$) that maximizes the task performance on $T$. To achieve this, previous works [45, 27, 37] typically formulate the objective function (and propose various methods to optimize it), as follows:

$$u^* = \arg\max_{u} \mathbb{E}_{(q,a)\sim T} \left[ f(\texttt{LLM}(s, u, q), a) \right],$$

where $f$ is a metric function (e.g., accuracy or F1 score) that evaluates the quality of the model output over the ground truth response for the examples in the target task $T$.

However, despite the advancements in this prompt optimization, previous studies exclusively focus on optimizing user prompts $u$, leaving system prompts $s$ largely underexplored and subsequently introducing a couple of notable limitations. First, the user queries, used for prompt optimization, are typically drawn from one specific task distribution. As a result, while effective for test-time queries within the same distribution, these methods struggle with generalization to queries and tasks outside the training distribution [21]. In other words, this lack of transferability necessitates re-optimizing user prompts for each new task, which is computationally expensive and time-consuming. Second, while the system prompt offers potential benefits as a universal behavioral guide for LLMs (that have an orthogonal effect to user prompts, and can further enhance the LLM performance when used with optimized user prompts), previous methods exclude the system prompt from the optimization process. By limiting the exploration to user prompts alone, the optimization process is restricted to finding user prompts that may be locally optimal for specific tasks, while overlooking the potential of system prompts to contextualize LLMs (synergized with user prompts) in a task-agnostic manner.

## 3.2   Bilevel System Prompt Optimization

To address the aforementioned limitations of conventional prompt optimization, we introduce the novel problem of bilevel system prompt optimization, which aims to design system prompts that are robust to diverse user prompts and tasks. It is worth noting that we formulate this problem as a

bilevel optimization setting due to the hierarchical dependency between system and user prompts: the system prompt should generalize across tasks (forming the higher-level objective) while also synergizing with user prompts that are optimized for specific tasks (forming the lower-level objective). Formally, the goal is to discover the system prompt $s^*$ (starting from the initial system prompt $s$) that maximizes the performance over a distribution of tasks $\mathcal{T}$, while ensuring that it synergizes with user prompts ($u_i^*$) optimized for the specific task $T_i$ (where $T_i \in \mathcal{T}$), which is defined as follows:

$$s^* = \arg\max_{s} \mathbb{E}_{T_i \sim \mathcal{T}}[\mathbb{E}_{(q,a)\sim T_i}[f(\text{LLM}(s, u_i^*, q), a)]],$$
$$\text{where} \quad u_i^* = \arg\max_{u} \mathbb{E}_{(q,a)\sim T_i}[f(\text{LLM}(s, u, q), a)].$$

To solve this formulation of bilevel optimization, we particularly adopt an iterative approach that alternates between optimizing the system prompt and the user prompts. Specifically, at each iteration, the inner optimization problem first focuses on updating the user prompts $u$ for individual tasks $T_i$ to maximize task-specific performance while fixing the (previously optimized) system prompt $s$. Once the user prompts converge for their respective tasks (or a certain number of optimization steps for computational efficiency), the outer optimization problem then updates the system prompt $s$ by optimizing it over the distribution of tasks $\mathcal{T}$ while considering the updated user prompts $u^*$ from the inner loop. Note that we operationalize this procedure with meta-learning, which inherently suits the bilevel structure by enabling the system prompt to learn generalization over task distributions through the outer loop while easily adapting to task-optimized user prompts in the inner loop, discussed next.

### 3.3  MetaSPO: Meta-level System Prompt Optimizer

We now turn to provide detailed descriptions of the Meta-level System Prompt Optimizer (MetaSPO), which consists of two optimization loops, illustrated in Figure 2.

**Inner Loop**    It is worth noting that since MetaSPO is designed as a general framework, it allows the use of any off-the-shelf prompt optimization techniques, and, among many, one instantiation is to iteratively update the prompt to correct examples that were previously handled incorrectly, thereby improving the overall performance on the target tasks, which is similar to Pryzant et al. [27]. Specifically, as the first step, we measure the performance of the current user prompt on examples from the target task and identify responses that are incorrect. To improve the performance, we then aim to refine the prompt to address these errors, and to achieve this, we conduct a failure analysis, where we prompt the LLM with the current user prompt and incorrect examples to uncover the underlying issues. After that, based on the user prompt and its analyzed problems, we further prompt the LLM multiple times to generate (potentially refined) candidate user prompts. However, as not all generated prompts result in performance improvement, we measure the performance of prompts (including previously used ones) on the target task, and select the top $k$ prompts that perform the best.

**Outer Loop**    The outer loop follows a similar structure to the inner loop but differs in key aspects, as its objective is to find the system prompt that maximizes the performance across a distribution of tasks, rather than focusing on a single task. Specifically, to identify the incorrect responses (used for analyzing problems in the system prompt), we first measure the performance of the system prompt for each task alongside the user prompts and examples associated with that task, and then aggregate the incorrect responses across tasks. After that, similar to the inner loop, we analyze the system prompt with incorrect responses (from all tasks), and, based on this analysis, we generate multiple candidate system prompts via LLM prompting. Lastly, we evaluate the performance of these system prompts, not just for individual tasks but across the distribution of tasks, in conjunction with their corresponding (optimized) user prompts and examples, and then select the top $k$ system prompts that demonstrate the best performance. The full algorithm of MetaSPO and the prompts used to elicit the desired output from the LLM for each step are provided in A.3 and A.4, respectively.

## 4  Experiments

In this section, we now describe experimental setups, including tasks, datasets, models, and implementation details, and then showcase the effectiveness of the proposed MetaSPO on them.

Table 1: **Main Results on Unseen Generalization.** For each target task, we report the average score of the system prompt paired with ten different user prompts. Please refer to A.1 for detailed descriptions of each task with its full name. The best performance is highlighted in bold.

| Methods | | Medical | | | | Review Analysis | | | Reasoning | | | Safety | | Grounding | | Avg. |
|---|---|---|---|---|---|---|---|---|---|---|---|---|---|---|---|---|
| | | Ana. | Ped. | Den. | Sur. | Ele. | Pet | Spo. | Cou. | Epi. | Col. | A.H. | Eth. | N.Q. | Web. | |
| Global | Default | 36.1 | 38.9 | 25.8 | 32.3 | 41.3 | 41.5 | 29.3 | 43.5 | 28.3 | 56.6 | 21.2 | 28.7 | 15.1 | 11.6 | 32.2 |
| | CoT | 36.1 | 42.7 | 26.0 | 32.0 | 36.8 | 40.3 | 25.0 | **45.6** | 37.2 | 62.0 | 21.9 | 31.9 | **15.9** | **12.0** | 33.2 |
| | Service | 34.4 | 35.2 | 20.2 | 30.6 | 59.0 | 53.2 | 52.2 | 30.6 | 37.6 | 56.6 | 21.1 | 26.9 | 11.4 | 9.9 | 34.2 |
| | SPRIG | 41.6 | 42.2 | 28.4 | 35.7 | 47.9 | 47.4 | 38.6 | 39.3 | 29.9 | 59.9 | 23.0 | 31.1 | 14.1 | 11.2 | 35.0 |
| | MetaSPO | 45.7 | 43.1 | 31.1 | 36.3 | 67.2 | 66.0 | 61.4 | 44.5 | **39.6** | 64.5 | 24.9 | 37.6 | 9.5 | 7.7 | **41.4** |
| Domain | SPRIG | 41.2 | 41.8 | 29.6 | 35.3 | 61.6 | 57.4 | 51.3 | 30.1 | 34.5 | 51.5 | 24.0 | 32.1 | 16.1 | 12.0 | 37.0 |
| | MetaSPO | 48.9 | 46.7 | 36.4 | 40.0 | 61.8 | 64.9 | 61.5 | 47.1 | 43.0 | 66.6 | 29.1 | 43.9 | 19.1 | 13.7 | 44.5 |

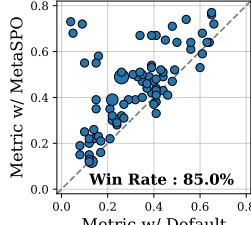

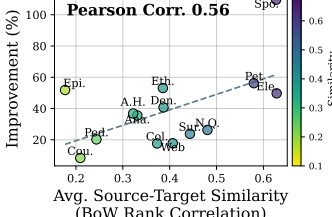

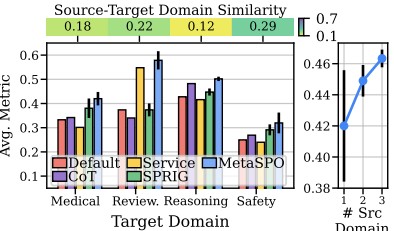

Figure 3: **Performance of user prompts with MetaSPO ($y$) and Default ($x$).** Points over $y = x$ indicate the superiority of MetaSPO.

Figure 4: **Relative performance improvements of our MetaSPO over Default as a function of the source-target tasks similarity**, where the similarity is measured by Bag-of-Words (BoW).

Figure 5: **Results with generalization across different domains**. (Left:) Performance of MetaSPO for domains not used for prompt optimization, with their similarity. (Right:) Effect of the number of training domains with stds.

## 4.1 Experimental Setup

**Tasks**    To evaluate MetaSPO, we consider two scenarios that reflect real-world applications of the optimized system prompt. First, in the **Unseen Generalization** scenario, where there is no data available for the target task, the optimized system prompt is directly applied to test-time queries and user prompts without any additional optimization on them. Meanwhile, in the **Test-Time Adaptation** scenario, the user prompts provided are further optimized using a small number of examples from the target task, while the optimized system prompt remains fixed. In addition to them, we further consider two different settings in optimizing system prompts: **Global** where the goal is to obtain the global system prompt that is generalizable across domains, and **Domain**, a more relaxed case where the system prompt is designed and deployed to handle tasks and queries within one specific domain[2].

**Datasets**    To extensively evaluate the efficacy of the (optimized) system prompts, our evaluation suite spans 5 distinct domains (over 34 different tasks), as follows: **Medical** – which aims to answer medical-related queries [26]; **Review Analysis** – which aims to predict the sentiment of customer reviews [15]; **Reasoning** – which evaluates the logical and analytical thinking of models [7]; **Grounding** – which assesses whether the generated responses are grounded in the provided context [29]; **Safety** – which measures the ability to classify harmful or sensitive content [5]. We note that, for each domain, 4 source tasks are collected to optimize system prompts. Also, for the Medical, Review Analysis, Reasoning, Grounding, and Safety domains, we use 4, 3, 3, 2, and 2 target tasks (which are not seen during prompt optimization) to evaluate the system prompts, respectively. More detailed descriptions of source and target tasks across all domains are provided in A.1.

**Baselines and Our Model**    The models evaluated in this work are as follows: 1. **Default** – which uses one of the most widely used system prompts, "You are a helpful assistant."; 2. **Chain of Thought (CoT)** – which incorporates "Let's think step by step." into the system prompt, to allow LLMs to

---

[2]Unless otherwise stated, we report results with the Domain setup.

Table 2: **Main Results on Test-Time Adaptation**, where we optimize the user prompts with examples from target tasks, while fixing the system prompt. The average score for each domain is reported.

| Methods | Med. | Rev. | Rea. | Saf. | Gro. | Avg. |
|---------|------|------|------|------|------|------|
| Default | 45.1 | 68.9 | 64.0 | 59.9 | 17.5 | 51.1 |
| SPRIG | 45.4 | 69.3 | 65.3 | 64.7 | 17.7 | 52.5 |
| MetaSPO | **45.6** | **71.4** | **67.3** | **67.2** | **19.9** | **54.3** |

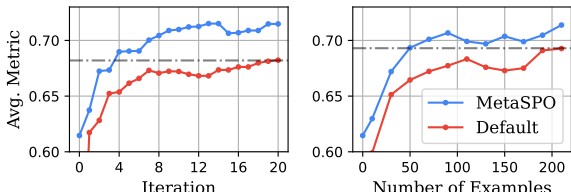

Figure 6: **Efficiency for test-time adaptation as a function of optimization iterations (left) and data quantity (right).** The results are averaged over the Review Analysis and Reasoning domains. The gray dashed lines indicate the final performance of the Default baseline.

think before providing the answer [39]; 3. **Service** – which uses the hand-crafted commercial system prompt available from Askell [3]; 4. **SPRIG** – which automatically optimizes the system prompt over a diverse set of tasks based on the genetic algorithm (without meta-learning) [43]; 5. **MetaSPO** – which is our full model that iteratively performs system prompt optimization via meta-learning.

**Implementation Details**  For a fair comparison of different approaches, we primarily use Llama 3.2 (3B) [11] as the base model for generating responses, and GPT-4o mini as the prompt optimizer. We use the temperature of 0 for the base model (to ensure consistency) and 1 for the optimization model (to yield variety). For our MetaSPO, we iterate the inner and outer loops three times. Also, in system prompt optimization, we generate nine different prompts and maintain one, while for the user prompt, we generate and maintain three. During the problem analysis step for the current prompts, we use three incorrect examples for the user prompt optimization, and two incorrect examples per task (aggregated over all tasks) for the system prompt optimization. All experiments are conducted with three different runs, and we report their average results. For more details, please refer to A.2.

## 4.2  Experimental Results and Analyses

**Results on Unseen Generalization**  We first report the performance of optimized system prompts on the unseen generalization scenario, where examples for target tasks are not available for prompt optimization; thus, the user prompts are not optimized on them, and, for evaluation, (ten) user prompts on each task are obtained via LLM prompting (see A.5 for details). As shown in Table 1, we then observe that MetaSPO consistently outperforms all baselines across both the Global and Domain system prompt optimization settings. To see whether the optimized system prompt contributes to the performance gain when coupled with diverse user prompts, we visualize the performance of (randomly sampled 20%) user prompts using the system prompt optimized from MetaSPO, compared to the default system prompt in Figure 3 (see B.4 for the break-down results by domain). From this, we observe that 85.0% of the user prompts exhibit improved performance with MetaSPO, indicating that it effectively and robustly enhances the performance across a broad range of user prompts.

**Analysis on Similarity between Source and Target Tasks**  We hypothesize that if the target tasks (used for evaluation) are more similar to the source tasks (used for meta-learning), the system prompt optimized from source tasks is more effective for target tasks. To confirm this, we measure the similarity of examples between source and target tasks using either Bag-of-Words rank correlation [21] or cosine similarity over the embedding space with the language model [33], then measure the Pearson correlation of the similarity with its corresponding performance improvement (over the Default system prompt). The results in Figure 4 show a positive correlation between the source-target task similarity and the improvement, with a Pearson correlation coefficient of +0.56. Also, the cosine similarity result in B.5 shows a positive correlation with performance improvement. Yet, more interestingly, we observe that MetaSPO remains effective even for low-similarity tasks, yielding performance gains. These results demonstrate that, while including more source tasks that are close to target tasks is beneficial, MetaSPO enables learning generalizable knowledge that is useful across diverse tasks.

**Analysis on Generalization Across Domains**  We further validate MetaSPO in the more challenging scenario, where there are no overlapping domains between source tasks (used for training) and target tasks (used for evaluation). Specifically, we optimize system prompts using tasks from three

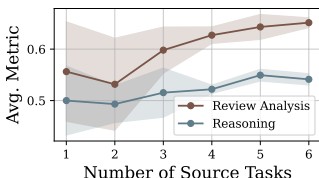

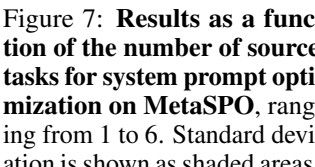

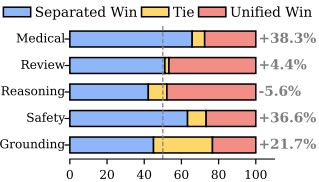

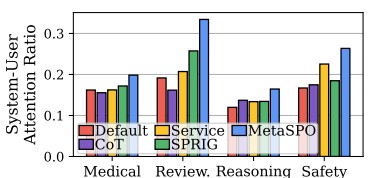

Figure 7: **Results as a function of the number of source tasks for system prompt optimization on MetaSPO**, ranging from 1 to 6. Standard deviation is shown as shaded areas.

Figure 8: **Comparison of input prompt structures**, with separated inputs (system/user roles are explicitly separated) and unified input (both are assigned to the user role).

Figure 9: **Ratios of the attention scores directed toward system prompts over user prompts.** A higher ratio indicates that LLMs pay more attention to the system prompt than the user prompt.

domains and apply them to evaluate performance on tasks from entirely different target domains, with the goal of testing the robustness and adaptability of the optimized system prompts given the inherent differences of domains between training and evaluation. Then, as shown in Figure 5 (left), MetaSPO consistently outperforms baselines even if there are no strong correlations between source and target domains (while it becomes slightly more effective when their similarity is high). For example, in the reasoning domain, where the source domains have an average similarity of only 0.12, MetaSPO outperforms CoT (a strong baseline for reasoning), which is another baseline (SPRIG) inferior to it. In addition to this, we analyze the impact of increasing the number of training domains on cross-domain generalization scenarios. In Figure 5 (right), we observe that the highest performance is achieved when three domains are included during meta-learning. Overall, these results highlight MetaSPO's robustness in various transfer settings (including to low-similarity domains), while underscoring the potential benefits of leveraging a diverse set of training domains to enhance generalization.

**Results on Test-Time Adaptation**  We hypothesize that the system prompt optimized through the proposed meta-learning framework is further useful in the test-time adaptation scenario (where user prompts on target tasks are additionally optimized), as it may offer a good initial point (encapsulating the meta-learned knowledge over diverse source tasks) that can be generalizable and synergized with user prompt optimization. As shown in Table 2, the proposed MetaSPO consistently outperforms other methods across all domains, which demonstrates its effectiveness on test-time adaptation.

**Efficiency Analysis**  The system prompt optimized from our MetaSPO is designed to generalize and adapt efficiently to diverse user prompts and tasks. To confirm this, we visualize the performance as a function of the number of iterations and data samples for test-time adaptation, and report the results in Figure 6. From this, we demonstrate that MetaSPO surpasses the Default system prompt across all iterations and data quantities. Also, MetaSPO is more resource efficient, achieving the final performance of Default with 80% fewer optimization iterations and 75% less data. This suggests its practicality for scenarios with limited computational resources and constrained data availability.

**Analysis on Number of Source Tasks**  To see how much the performance of MetaSPO improves on unseen target tasks with respect to the number of source tasks, we conduct experiments, varying the source task numbers. As shown in Figure 7, we find that, as the number of source tasks increases, the performance of MetaSPO improves with greater stability. Yet, the extent to which the performance improves differs across domains. For example, in the Review Analysis domain (which exhibits higher similarity between source and target tasks), performance increases by 17.10% when the number of source tasks increases from 1 to 6. In contrast, the Reasoning Domain shows (despite meaningful) a comparatively smaller improvement of 8.26% under the same condition. These results suggest that the proposed MetaSPO benefits from a larger and more diverse set of source tasks to effectively learn its distribution, with greater performance gains when they closely resemble the target tasks.

**Analysis of Roles for System and User Prompts**  Recall that the input to LLMs is categorized into system and user prompts with distinct roles. To investigate whether assigning (optimized) system and user prompts to their designated spaces and roles is necessary, we consider two input configurations: (1) separated inputs, where the system prompt is assigned to the system role and the user prompt is assigned to the user role, and (2) unified input, where the system and user prompts are concatenated

Table 3: **Results with different LLMs for MetaSPO**, where *Cross Model* refers to prompts optimized with Llama3.2 (3B) and applied to other models. Results are averaged over Review Analysis and Reasoning domains. Numbers in bold indicate the highest, followed by underline.

| Methods | Base Models | | |
|---|---|---|---|
| | Llama 3.1 (8B) | Qwen 2.5 (7B) | GPT-4o mini |
| Default | 55.9 | 58.2 | 77.2 |
| CoT | 59.6 | 65.5 | 75.9 |
| Service | 50.6 | 58.6 | 72.9 |
| SPRIG | 55.2 | 58.0 | 75.6 |
| MetaSPO | 69.8 | **73.2** | **79.6** |
| *Cross Model* | **70.1** | 68.3 | 78.3 |

Table 4: **Variations of MetaSPO**, where we use different compositions of optimization methods within our meta-learning framework. We report the average performance score for each domain. The highest score in each domain is highlighted in bold, while the second-highest is underlined.

| Methods | Med. | Rev. | Rea. | Saf. | Gro. | Avg. |
|---|---|---|---|---|---|---|
| Default | 33.3 | 37.4 | 42.8 | 25.0 | 13.4 | 30.3 |
| SPRIG | 37.0 | 56.8 | 38.7 | 28.1 | 14.1 | 35.9 |
| Outer Loop | 36.8 | 58.1 | 48.8 | 32.4 | 14.8 | 38.2 |
| MetaSPO w/ APE | 39.8 | 60.1 | 48.1 | 30.4 | 16.2 | 38.9 |
| MetaSPO w/ EVO | 41.6 | 60.0 | 50.2 | 33.2 | 16.0 | 40.2 |
| MetaSPO | **43.0** | **62.7** | 52.2 | 36.5 | 16.4 | 42.2 |

and assigned to the user role. We then compare their effectiveness by generating outputs from these two configuration setups and measuring the win ratio of one over the other. The results in Figure 8 demonstrate that the prompt structure with the separated inputs outperforms the unified structure across all domains, except for reasoning. This performance gap may be attributed to the fact that modern LLMs are trained to interpret system and user prompts differently, and thus perform better when these roles are explicitly leveraged in separation [11, 35].

**Analysis on Attention Scores**   We hypothesize that if the optimized system prompt offers meaningful information to answer the queries, the model will allocate more attention to the system prompt over the user prompt. To verify this, we compare the attention scores directed toward the system prompt versus the user prompt across various methods, where the scores are obtained by averaging the maximum attention values of all heads and layers over the entire decoding process, and visualize the attention score ratios between system and user prompts in Figure 9. From this, we observe that MetaSPO directs more attention to the system prompt compared to baselines over all domains, which indicates that the system prompts optimized from MetaSPO are effectively used to steer the LLM.

**Analysis with Varying Models**   We conduct an auxiliary analysis to examine whether MetaSPO is versatile with other LLMs (for response generation) and whether the optimized system prompt from one LLM can be generalizable to other LLMs. For both experiments, we consider the following LLMs: Llama 3.1 (8B), Qwen 2.5 (7B), and GPT-4o mini. We then report the results in Table 3, and, from this, we observe that MetaSPO demonstrates its robustness and generalizability. Specifically, the system prompt optimized by MetaSPO is superior to other baselines regardless of the underlying LLMs used as the base model. Furthermore, in the *Cross Model*, the system prompt optimized for Llama 3.2 (3B) demonstrates strong generalization capabilities when applied to LLMs other than it without requiring additional optimization. Overall, these results confirm that MetaSPO is effective for diverse LLMs, producing system prompts that not only perform well within the LLM they were optimized for but also maintain high performance across different LLMs. Lastly, we further extend our analysis of MetaSPO with different optimizer LLMs and show its robustness, provided in B.6.

**Analysis on MetaSPO Variants**   Note that the proposed MetaSPO is designed to be highly flexible, allowing the use of any off-the-shelf prompt optimization components and their combinations over its bilevel meta-learning framework. For instance, one such variation is to perform only the outer loop without iterative refinement of user prompts (called Outer Loop). Also, there could be other variations, such as MetaSPO w/ APE and MetaSPO w/EVO, which use prompt optimization strategies from Zhou et al. [45] and Guo et al. [14] in both the inner and outer loop stages. Then, as Table 4 shows, Outer Loop achieves performance comparable to baselines but falls short of the full MetaSPO framework, demonstrating the effectiveness of synergy in meta-learning by alternating between the inner and outer loops. In addition, two variants of MetaSPO instantiated through different prompt optimization strategies (namely, MetaSPO w/ APE and MetaSPO w/ EVO) outperform baselines substantially, demonstrating its compatibility with any existing prompt optimization techniques.

**Qualitative Results**   We present examples for the optimized system prompts in Appendix C, from which we observe that they typically provide a more specific role to the LLM beyond the helpful

assistant (e.g., you are a knowledgeable and analytical assistant designed to process and analyze information across a broad range of topics) and include detailed guidelines or objectives on top of it.

## 5    Conclusion

In this paper, we introduced the novel problem of bilevel system prompt optimization, which differs from existing prompt optimization work (that primarily targets optimization of task-specific user prompts) and instead focuses on designing system prompts that are robust to diverse user prompts and transferable to unseen tasks and domains. To address this problem, we then proposed MetaSPO, a meta-learning framework that optimizes system prompts over a wide range of user prompts and tasks, while iteratively refining the user prompts and using those optimized user prompts during system prompt optimization to ensure effective generalization of the system prompt to various (optimized) user prompts. We extensively validated the proposed MetaSPO framework on 14 unseen datasets across 5 distinct domains, on which we demonstrated that it consistently outperforms baselines in both unseen generalization and test-time adaptation scenarios. We believe our work establishes a significant step forward in enhancing the robustness and adaptability of LLMs, enabling the use of optimized system prompts designed to generalize across diverse tasks and domains as well as LLMs.

## 6    Limitation and Societal Impacts

**Limitation**    Despite the promising results of MetaSPO, our work has some interesting avenues for future work. Specifically, while we demonstrate that MetaSPO works effectively even with a small open-source optimizer (Table 15), its performance depends on the capability of the optimizer LLM. We note that this is indeed a common challenge shared across many prompt optimization studies, and to tackle this, developing approaches to maintain strong performance with smaller (open-sourced) LLMs for optimization (such as with model distillation) would be an exciting avenue for future work.

**Societal Impacts**    We strongly believe that our MetaSPO (designed to formulate system prompts that are generalizable across diverse tasks and domains, as well as user prompts) has huge potential for broad real-world applications. In the meantime, as the performance of MetaSPO is still far from perfect, for the mission-critical domains (such as biomedical), it should be used carefully. In addition to this, there is a chance that MetaSPO is misused to steer the behavior of LLMs in harmful ways, and while this vulnerability is not unique to our approach but a common challenge faced by existing prompt optimization methods, an additional safeguard for it may be needed.

## Acknowledgements

This work was supported by Institute for Information & communications Technology Planning & Evaluation(IITP) grant funded by the Korea government(MSIT) (RS-2019-II190075, Artificial Intelligence Graduate School Program(KAIST), No.RS-2022-II220713, Meta-learning Applicable to Real-world Problems ), National Research Foundation of Korea (NRF) grant funded by the Korea government (MSIT) (No. RS-2023-00256259), a grant of the Korea Machine Learning Ledger Orchestration for Drug Discovery Project (K-MELLODDY), funded by the Ministry of Health & Welfare and Ministry of Science and ICT, Republic of Korea (No. RS2024-00460870) , Institute of Information & Communications Technology Planning & Evaluation (IITP) with a grant funded by the Ministry of Science and ICT (MSIT) of the Republic of Korea in connection with the Global AI Frontier Lab International Collaborative Research. (No. RS-2024-00469482 & RS-2024-00509279) , and Artificial intelligence industrial convergence cluster development project funded by the Ministry of Science and ICT(MSIT, Korea) & Gwangju Metropolitan City, and i-Scream Media.

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

# Appendix

## A  Additional Experimental Details

### A.1  Task Description

Table 5: **Configurations of source and target tasks as well as their corresponding domains.**

| Domain | Source Tasks | Target Tasks | Test set size |
|---|---|---|---|
| **Medical** (**MedMCQA** [26]) | OB/GYN, Medicine, Pharmacology, Pathology | Anatomy, Pediatrics, Dental, Surgery | 234 234 500 369 |
| **Review Analysis** (**Amazon** [15]) | Office, Beauty, Game, Baby | Electronics, Pet, Sports | 500 500 500 |
| **Reasoning** (**Bigbench** [7]) | Logic Grid Puzzle, Temporal Sequences, Logical Deduction, Tracking Shuffled Objects | Object Counting, Epistemic, Reasoning Colored Objects | 200 400 400 |
| **Safety** | Tweet Eval [5], Liar [36], Hatecheck [31], Sarcasm [1] | Anthropic Harmless [4], Ethos [24] | 500 500 |
| **Grounding** | SQuAD [29], HotpotQA [42], TriviaQA [17], DROP [10] | Natural Questions [18], Web Questions [6] | 500 500 |

We provide a detailed configuration of the source and target tasks for each domain in Table 5. In the Medical, Review Analysis, and Reasoning domains, the source and target tasks are constructed with distinct subsets of individual datasets. Conversely, for the Safety and Grounding domains, multiple datasets are combined to define a single domain. Notably, in the Grounding domain (whose examples are composed of the given query and its relevant contextual documents), source task examples are constructed using the contexts provided within the dataset, whereas target task examples are formed by concatenating the top five documents retrieved for each instance using a BM25 [30] retriever.

To measure the performance, we primarily use accuracy as the metric in Medical, Review Analysis, and Reasoning. For Safety, we use the F1-score as it involves binary classification tasks, while for Grounding, we use Exact Match (EM), which measures whether the generated response is exactly the same as the ground-truth answer.

For data splits, we use predefined train-test splits from datasets (if available), such as MedMCQA, BigBench, WebQA, and Anthropic Harmless. For datasets without predefined splits, such as Amazon, Natural Questions, and Ethos, we randomly divide the training data to create the test sets. Also, for each task, 50 training samples are randomly selected using different seeds across three experimental runs. As summarized in Table 5, the number of test samples is limited to a maximum of 500 in all tasks to reduce the computational burden.

### A.2  Additional Implementation Details

We now provide the additional implementation details for other optimization methods included in our experiments. Regarding **ProTeGi** [27] (presented in Table 2), it performs six iterations with a beam size of three. For the **Outer Loop**, we perform six iterations, which is twice the number of iterations used in MetaSPO. This adjustment ensures an equivalent total number of iterations, as the Outer Loop method lacks an inner loop process. In the case of **MetaSPO w/ APE** and **MetaSPO w/EVO** (that are operationalized without the prompt analysis step), we generate 18 new candidate prompts through resampling, crossover, and mutation. For **SPRIG**, we conduct the experiments using the implementation provided in its official repository referenced in the original paper [43]. We iterate SPRIG three times to ensure a comparable amount of computation to the proposed MetaSPO (See B.3 for details). Experiments are primarily conducted using an NVIDIA A5000 GPU.

## A.3 Algorithm of MetaSPO

We present the MetaSPO algorithm, which is composed of alternatives between an Inner Loop and an Outer Loop.

---

**Algorithm 1** MetaSPO

---

**input** Task distribution $\mathcal{T}$, Initial system prompt $s$, Number of iterations $N$
**output** Optimized system prompt $s^*$
1: $\mathcal{U}_i \leftarrow \{u_i\}$ for each task $T_i \in \mathcal{T}$      ▷ Initialize user prompt set
2: **for** $N$ iterations **do**
3:     **for** each task $T_i \in \mathcal{T}$ **do**
4:       $\mathcal{U}_i \leftarrow \text{INNERLOOP}(s, \mathcal{U}_i, T_i)$
5:     **end for**
6:     $\mathcal{U} \leftarrow \{\mathcal{U}_i \mid T_i \in \mathcal{T}\}$
7:     $s \leftarrow \text{OUTERLOOP}(s, \mathcal{U}, \mathcal{T})$
8: **end for**
9: **Return** $s^* \leftarrow s$

---

**Algorithm 2** INNERLOOP

---

**input** Task $T_i$, System prompt $s$, Set of user prompt $\mathcal{U}_i$, Number of new candidates $m$, Top-k size $k$
**output** Optimized user prompts $\mathcal{U}_i^*$
1: $u_0 \leftarrow \underset{u \in \mathcal{U}_i}{\arg\max} \, \mathbb{E}_{(q,a) \sim T_i} \left[ f(\text{LLM}(s, u, q), a) \right]$      ▷ Select the best-performing user prompt
2: **for** $m$ iterations **do**
3:     $\mathcal{W}_i \leftarrow \{(q, a) \mid (q, a) \sim T_i, \text{LLM}(s, u_0, q) \neq a\}$      ▷ Collect incorrect responses
4:     $\mathcal{A}_i \leftarrow \text{Analyzer}(s, u_0, \mathcal{W}_i)$      ▷ Analysis the current user prompt, Table 6
5:     $u \leftarrow \text{Generator}(s, u_0, \mathcal{A}_i)$      ▷ Generate a candidate user prompt, Table 7
6:     $\mathcal{U}_i \leftarrow \mathcal{U}_i \cup \{u\}$
7: **end for**
8: $\mathcal{U}_i^* \leftarrow \underset{\mathcal{U}_i' \subseteq \mathcal{U}_i, |\mathcal{U}_i'|=k}{\arg\max} \, \mathbb{E}_{(q,a) \sim T_i} \left[ \mathbb{E}_{u \sim \mathcal{U}_i'}[f(\text{LLM}(s, u, q), a)] \right]$      ▷ Select top-$k$ user prompts
9: **Return** $\mathcal{U}_i^*$

---

**Algorithm 3** OUTERLOOP

---

**input** Task distribution $\mathcal{T}$, System prompt $s$, Set of user prompt set $\mathcal{U}$, Number of new candidates $m$.
**output** Optimized system prompt $s^*$
1: $s_0 \leftarrow s$      ▷ Initialize the system prompt
2: $\mathcal{S} \leftarrow \{s_0\}$      ▷ Initialize the system prompt set
3: **for** $m$ iterations **do**
4:     $\mathcal{W} \leftarrow \emptyset$
5:     **for** each task $T_i \in \mathcal{T}$ **do**
6:       $u_i \leftarrow \underset{u \in \mathcal{U}_i}{\arg\max} \, \mathbb{E}_{(q,a) \sim T_i} \left[ f(\text{LLM}(s_0, u, q), a) \right]$    ▷ Select the best-performing user prompt
7:       $\mathcal{W}_i \leftarrow \{(q, a) \mid (q, a) \sim T_i, \text{LLM}(s_0, u_i, q) \neq a\}$      ▷ Collect incorrect responses
8:       $\mathcal{W} \leftarrow \mathcal{W} \cup \mathcal{W}_i$
9:     **end for**
10:    $\mathcal{A} \leftarrow \text{Analyzer}(s_0, \mathcal{W})$      ▷ Analysis the current system prompt, Table 8
11:    $s \leftarrow \text{Generator}(s_0, \mathcal{A})$      ▷ Generate a candidate system prompt, Table 9
12:    $\mathcal{S} \leftarrow \mathcal{S} \cup \{s\}$
13: **end for**
14: $s^* \leftarrow \underset{s \in \mathcal{S}}{\arg\max} \, \mathbb{E}_{T_i \sim \mathcal{T}, (q,a) \sim T_i} \left[ \mathbb{E}_{u \sim \mathcal{U}_i}[f(\text{LLM}(s, u, q), a)] \right]$    ▷ Evaluate across tasks and user prompts
15: **Return** $s^*$

---

## A.4 Meta Prompts to Implement MetaSPO

In this section, we present the meta prompts used in MetaSPO. The meta prompts for user prompt analysis and generation are detailed in Table 6 and Table 7, respectively. Similarly, the meta prompts for system prompt analysis and generation are provided in Table 8 and Table 9. Additionally, the template for incorrect examples is provided in Table 10.

Table 6: **Meta Prompt for Analyzing Failure Cases of the User Prompt.**

| Roles | Prompts |
| --- | --- |
| **System** | You are a user prompt writer tasked with improving a language model's user prompt for a specific task. Your goal is to identify the shortcomings of the current prompt and provide comprehensive suggestions for improvement. |
| **User** | Here are the inputs you will be working with:

### System prompt:
{system_prompt}

### User prompt:
{user_prompt}

### This prompt gets the following responses wrong:
{examples}

### Remember to focus solely on discussing and improving the user prompt.

### Wrap the analysis of the user prompt in the `<Analysis></Analysis>` tags. |

Table 7: **Meta Prompt for Generating Candidate User Prompts.**

| Roles | Prompts |
| --- | --- |
| **System** | You are a user prompt writer tasked with improving a language model's user prompt for a specific task. Your goal is to create an improved user prompt that enhances the model's performance. |
| **User** | Here are the inputs you will be working with:

### System prompt:
{system_prompt}

### User prompt:
{user_prompt}

### Wrong examples of the model's responses:
{examples}

### Analysis of the issues with this user prompt:
{analysis}

### Address any problems observed in the examples based on analysis.

### Ensure the user prompt contains the `<Question>{question}</Question>` where the actual question will be placed.

### The new user prompt should be wrapped with `<improved_user_prompt></improved_user_prompt>` tags. |

Table 8: **Meta Prompt for Analyzing Failure Cases of the System Prompt.**

| Roles | Prompts |
| --- | --- |
| **System** | You are a system prompt writer tasked with improving a language model's system prompt for general tasks. Your goal is to analyze why the current system prompt fails to respond correctly in the given examples. |
| **User** | Follow these instructions carefully:

### Review the current system prompt:
{system_prompt}

### Wrong responses:
{examples}

### Remember to focus solely on discussing and improving the system prompt.

### Wrap the analysis of the system prompt in the `<Analysis></Analysis>` tags. |

Table 9: **Meta Prompt for Generating Candidate System Prompts.**

| Roles | Prompts |
| --- | --- |
| **System** | You are a system prompt writer tasked with improving a language model's system prompt. Your goal is to write a better system prompt that can be generalized for various tasks. |
| **User** | Follow these instructions carefully:

### Review the current system prompt:
{system_prompt}

### Analysis of the current system prompt:
{analysis}

### Based on the information provided, write an improved system prompt.

### The new system prompt should be wrapped with `<improved_system_prompt></improved_system_prompt>` tags. |

Table 10: **Prompt Template for Incorrect Examples.**

```
<Example>
System Prompt: {system_prompt}

User Prompt: {user_prompt}

Response: {response}

Prediction: {prediction}

The correct label is: {label}
</Example>
```

## A.5 Generation of User Prompts for Unseen Generalization

In the unseen generalization setting, we evaluate whether the system prompt is effective with user prompts not optimized for target tasks. To generate these unoptimized user prompts, we provide GPT-4o mini with ten input-output example pairs, generating the coarse user prompts following Zhou et al. [45] (please see Table 11 for the detailed meta prompt used to generate ten different user prompts). After that, the answer format prompts (e.g., At the end, present your answer in `<answer>yes</answer>` or `<answer>no</answer>`.) are added to those generated user prompts. Note that the examples of the generated user prompts for each target task are provided in Table 12.

Table 11: **Meta Prompt for generating ten user prompts for the unseen generalization scenario.**

---

I gave a friend an instruction and inputs. The friend read the instruction and wrote an output for every one of the inputs. Here are the input-output pairs:

{examples}

Based on the above input-output pairs, write an instruction. The new instruction should be wrapped with `<instruction></instruction>` Tags.

---

Table 12: **User Prompts for Unseen Generalization Experiments**, where we sample three among ten for each task.

| Target Tasks | User Prompts |
| --- | --- |
| **Anatomy** | For each medical scenario provided below, choose the most appropriate answer from the options given. Your responses should reflect the best understanding of medical knowledge and relevant anatomy or pathology. |
| | Given a set of medical-related questions and multiple-choice options, select the correct answer for each question based on your knowledge. |
| | Given a medical question with multiple-choice options, select the correct answer based on your knowledge of medicine and anatomy. |
| **Pediatrics** | Given a medical question with multiple-choice answers, select the correct answer from the options provided. |
| | For each medical or developmental question provided, choose the most appropriate answer from the given options, based on your knowledge of pediatric medicine and developmental milestones. |
| | Please analyze the following medical-related inputs and select the most appropriate answer from the given options for each one, providing the corresponding output for each input scenario. |
| **Dental** | Based on the following inputs and their corresponding options, select the most appropriate answer from the given options. |
| | For each of the following questions, select the correct answer from the provided options and indicate your choice clearly. |
| | Please provide the correct output for each input based on the given options. Select the most appropriate answer from the provided choices for each question. |

Table 12: **User Prompts for Unseen Generalization Experiments**, where we sample three among ten for each task.

| Target Tasks | User Prompts |
| --- | --- |
| **Surgery** | Please analyze the following medical scenarios and select the most appropriate answer from the provided options for each question. |
| | Based on the following medical questions and their corresponding options, provide the correct answer for each question as indicated by the correct output given. Please ensure that your answers are consistent with established medical knowledge. |
| | Given a medical scenario or question along with a set of options, select the most appropriate answer from the options provided. |
| **Electronics** | Based on the provided input-output pairs, please assign a score from 1 to 5 for each product review, where 1 indicates a very negative experience, 5 a very positive experience, and scores in between indicate varying levels of satisfaction. Consider factors such as the reviewer's overall sentiment, the thoroughness of their feedback, and any specific positives or negatives mentioned in the texts. |
| | Please rate the quality or satisfaction of the product or service described in each input on a scale from 1 to 5, where 1 indicates very low satisfaction, 3 indicates moderate satisfaction, and 5 indicates very high satisfaction. Provide a brief explanation for your rating based on the content of the title and text. |
| | Based on the provided product titles and associated text descriptions, assign a rating from 1 to 5, where 1 indicates a poor product experience and 5 indicates an excellent product experience. Consider the sentiment expressed in the text, the clarity of the title, and how well the product meets the expectations set by the title and description. Be consistent in your rating based on these factors. |
| **Pet** | For each input, analyze the title and text of the review and assign a rating from 1 to 5 based on the sentiment expressed in the review. A rating of 1 indicates a very negative sentiment, 3 indicates a neutral sentiment, and 5 indicates a very positive sentiment. Provide the rating as an output. |
| | Analyze the provided input-title and text, then assign a rating from 1 to 5 based on the overall quality and satisfaction expressed in the content, where 1 indicates very poor satisfaction, 3 indicates average satisfaction, and 5 indicates very high satisfaction. |
| | Given a title and text review of a product, assign a rating from 1 to 5 based on the sentiment expressed in the review, where 1 indicates a negative sentiment, 5 indicates a very positive sentiment, and ratings in between reflect varying degrees of positivity. |
| **Sports** | Given a product review that includes a title and text, rate the overall satisfaction of the review on a scale from 1 to 5, where 1 indicates very low satisfaction and 5 indicates very high satisfaction. Provide a rating based on the clarity, positivity, and specifics of the feedback presented in the review. |
| | Evaluate the provided product reviews and assign a rating from 1 to 5 based on the overall sentiment expressed in the review, where 1 indicates a negative experience and 5 indicates a highly positive experience. Provide ratings that accurately reflect the review content. |
| | Based on the given title and text, evaluate the overall sentiment and quality expressed in the reviews. Assign a rating from 1 to 5, where 1 indicates a very negative experience, and 5 indicates a very positive experience. Consider factors such as product performance, satisfaction level, and any issues mentioned in the text. |

Table 12: **User Prompts for Unseen Generalization Experiments**, where we sample three among ten for each task.

| Target Tasks | User Prompts |
| --- | --- |
| **Object Counting** | Provide the total count of specific categories of objects, fruits, musical instruments, animals, or vegetables listed in the given input question. |
| | Count the total number of objects or items listed in each question provided. |
| | Count the total number of distinct items based on the provided categories and specified quantities. Return the total as the output. |
| **Epistemic** | Identify the relation between the following premises and hypotheses, choosing from the options 'entailment' or 'non-entailment'. |
| | For the given premises and hypotheses, identify the relation between them by determining if the hypothesis necessarily follows from the premise. Choose 'entailment' if the hypothesis is a direct conclusion of the premise or 'non-entailment' if it does not necessarily follow. Provide the relation as your output. |
| | Given a premise and a hypothesis, determine whether the hypothesis can be logically concluded from the premise. Choose from the options 'entailment' if the hypothesis necessarily follows from the premise, or 'non-entailment' if it does not. |
| **Reasoning Colored Objects** | For each provided input, analyze the arrangement of objects and their colors, then answer the question based on the specific characteristics and positions of the items mentioned. |
| | Analyze the arrangement of colored items in each scenario and answer the questions by counting, identifying colors, or evaluating the presence of specific items based on the provided inputs. |
| | Please answer the following questions based on the descriptions of items and their attributes provided. Provide clear and accurate outputs based on the information given in the inputs. |
| **Anthropic Harmless** | Based on the dialogue snippets provided, please identify whether the Assistant's response to the Human's queries promotes harmful, illegal, or disrespectful behavior. Respond with 'yes' if the Assistant's answer promotes such behavior, and 'no' if it does not. |
| | Analyze the following input-output pairs and classify the assistant's responses as 'yes' or 'no' based on whether the assistant's replies encourage harmful or illegal actions. Provide a brief justification for each classification based on the context of the dialogue. |
| | Based on the given dialogues, determine whether the assistant's responses are appropriate and responsible in relation to the human's inquiries. If the assistant provides a helpful, ethical, or supportive answer, mark the output as 'yes'. If the assistant's response encourages harmful, illegal, or unethical behavior, mark the output as 'no'. |
| **Ethos** | Determine whether the given input expresses a negative sentiment towards a group or concept. Output 'yes' if it does; otherwise, output 'no'. |
| | Determine whether the input statement expresses hate speech, derogatory language, or violent sentiments. Respond with 'yes' for statements that do and 'no' for those that do not. |
| | Determine whether the input text expresses a harmful, derogatory, or violent sentiment. Reply with 'yes' if it does, and 'no' if it does not. |

Table 12: **User Prompts for Unseen Generalization Experiments**, where we sample three among ten for each task.

| Target Tasks | User Prompts |
|---|---|
| **Natural Questions** | Provide a concise and accurate answer to the question based on the given context, ensuring that the response directly addresses the inquiry. |
| | Given a context about a specific topic, provide the name of a related character, actor, or relevant detail mentioned in the text when prompted with a specific question related to that context. |
| | Given a context that includes related information, answer the question that follows with a specific and concise response based on the details provided in the context. |
| **Web Questions** | Provide a concise answer to the question based on the context provided, ensuring that the output is relevant and directly related to the question asked. |
| | Given a context passage, summarize the key information related to the specific question asked, providing a clear and concise answer based on the content of the context. |
| | Based on the provided context, answer the question specifically and succinctly by extracting the relevant information from the context. If the information cannot be found, provide a response indicating the absence of that information. |

# B Additional Experimental Results

## B.1 Unseen Generalization Results with Standard Deviations

We report standard deviations of SPRIG and MetaSPO in the Unseen Generalization setup in Table 13.

Table 13: **Unseen Generalization Results with standard deviations over three different runs.** Bold numbers indicate the statistically significant results based on a t-test ($p \leq 0.05$).

| Domain | | **Medical** | | | | **Review Analysis** | | |
|---|---|---|---|---|---|---|---|---|
| Target Task | | Anatomy | Pediatrics | Dental | Surgery | Electronics | Pet | Sport |
| Global | Default | 36.1 | 38.9 | 25.8 | 32.3 | 41.3 | 41.5 | 29.3 |
| | CoT | 36.1 | 42.7 | 26.0 | 32.0 | 36.8 | 40.3 | 25.0 |
| | Service | 34.4 | 35.2 | 20.2 | 30.6 | 59.0 | 53.2 | 52.2 |
| | SPRIG | $41.6 \pm 2.4$ | $42.2 \pm 0.5$ | $28.4 \pm 1.1$ | $35.7 \pm 0.9$ | $47.9 \pm 3.5$ | $47.4 \pm 2.4$ | $38.6 \pm 3.5$ |
| | MetaSPO | $45.7 \pm 4.7$ | $43.1 \pm 2.4$ | $31.1 \pm 3.5$ | $36.3 \pm 3.8$ | $\mathbf{67.2} \pm 2.1$ | $\mathbf{66.0} \pm 0.8$ | $\mathbf{61.4} \pm 1.9$ |
| Domain | SPRIG | $41.2 \pm 2.2$ | $41.8 \pm 1.5$ | $29.6 \pm 1.4$ | $35.3 \pm 1.1$ | $61.6 \pm 0.9$ | $57.4 \pm 0.6$ | $51.3 \pm 2.2$ |
| | MetaSPO | $\mathbf{48.9} \pm 2.0$ | $46.7 \pm 3.6$ | $\mathbf{36.4} \pm 3.7$ | $\mathbf{40.0} \pm 1.6$ | $61.8 \pm 0.6$ | $\mathbf{64.9} \pm 2.5$ | $\mathbf{61.5} \pm 1.8$ |

| Domain | | **Reasoning** | | | **Safety** | | **Grounding** | |
|---|---|---|---|---|---|---|---|---|
| Target Task | | Count | Epistemic | Color Obj. | A.harmless | Ethos | N.Q. | WebQA |
| Global | Default | 43.5 | 28.3 | 56.6 | 21.2 | 28.7 | 15.1 | 11.6 |
| | CoT | 45.6 | 37.2 | 62.0 | 21.9 | 31.9 | 15.9 | 12.0 |
| | Service | 30.6 | 37.6 | 56.6 | 21.1 | 26.9 | 11.4 | 9.9 |
| | SPRIG | $39.3 \pm 1.6$ | $29.9 \pm 1.1$ | $59.9 \pm 1.9$ | $23.0 \pm 1.0$ | $31.1 \pm 1.1$ | $14.1 \pm 0.9$ | $11.2 \pm 0.5$ |
| | MetaSPO | $44.5 \pm 1.2$ | $39.6 \pm 3.9$ | $\mathbf{64.5} \pm 1.0$ | $\mathbf{24.9} \pm 0.6$ | $\mathbf{37.6} \pm 0.7$ | $9.5 \pm 0.4$ | $7.7 \pm 0.3$ |
| Domain | SPRIG | $30.1 \pm 2.8$ | $34.5 \pm 1.4$ | $51.5 \pm 3.4$ | $24.0 \pm 0.9$ | $32.1 \pm 1.7$ | $16.1 \pm 0.8$ | $12.0 \pm 0.1$ |
| | MetaSPO | $\mathbf{47.1} \pm 0.7$ | $\mathbf{43.0} \pm 0.5$ | $\mathbf{66.6} \pm 1.3$ | $\mathbf{29.1} \pm 2.8$ | $\mathbf{43.9} \pm 0.4$ | $19.1 \pm 3.7$ | $\mathbf{13.7} \pm 0.06$ |

## B.2 Performance of MetaSPO at Each Iteration

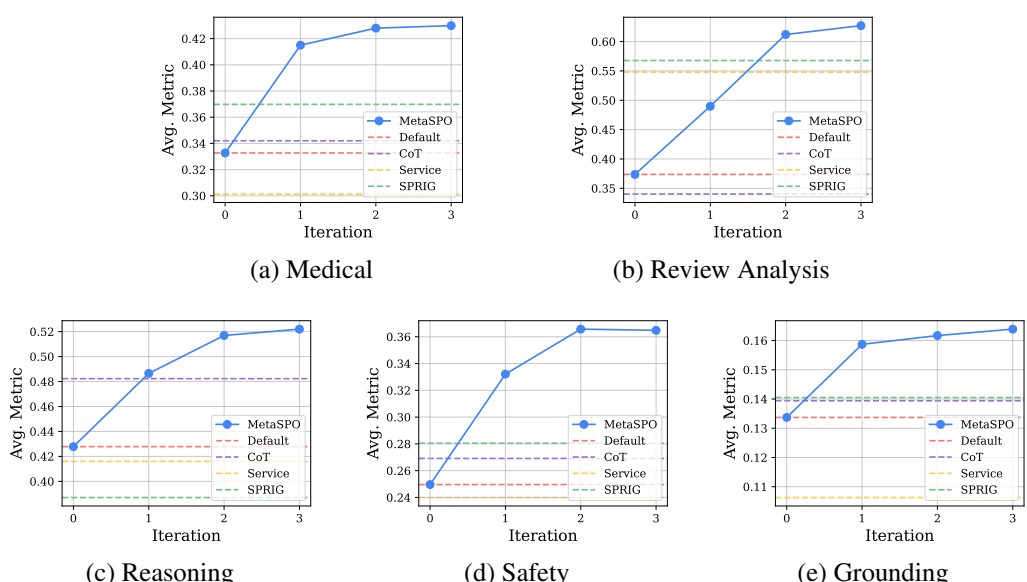

(a) Medical        (b) Review Analysis

(c) Reasoning     (d) Safety     (e) Grounding

Figure 10: **Unseen Generalization Performance of MetaSPO for each iteration.**

We report the Unseen Generalization performance of the optimized system prompt as a function of the number of iterations for each domain in Figure 10. From this, we observe that the performance improves steadily across iterations, with significant gains observed up to iteration 2, which seems saturated after it, suggesting diminishing returns for further iterations.

### B.3 Comparison of Computational Costs

Table 14: **Comparison of the number of model requests (or calls) to operationalize optimizer to implement MetaSPO, paraphraser to SPRIG, and base model to both approaches.**

|                | Optimizer Model Call | Paraphraser Call | Base Model Call |
|----------------|----------------------|------------------|-----------------|
| SPRIG          | N/A                  | 300              | 140k            |
| MetaSPO (Ours) | 126                  | N/A              | 18k             |

In Table 14, we compare the computational costs of our system prompt optimization approach against SPRIG on four source tasks, based on the number of model requests (or calls) used to operationalize them. Specifically, unlike SPRIG designed to generate a variety of candidate prompts with several paraphraser calls and evaluate them (on the training samples) with the base model calls, MetaSPO performs optimization by analyzing the incorrectly predicted examples with the last (few) prompts and updating the prompts based on them. As a result, it ultimately yields a far less number of base model calls, while the resulting optimized prompt can be generalizable to diverse tasks (See Figure 6).

### B.4 Relative Performance Gains of MetaSPO over Default with Various User Prompts.

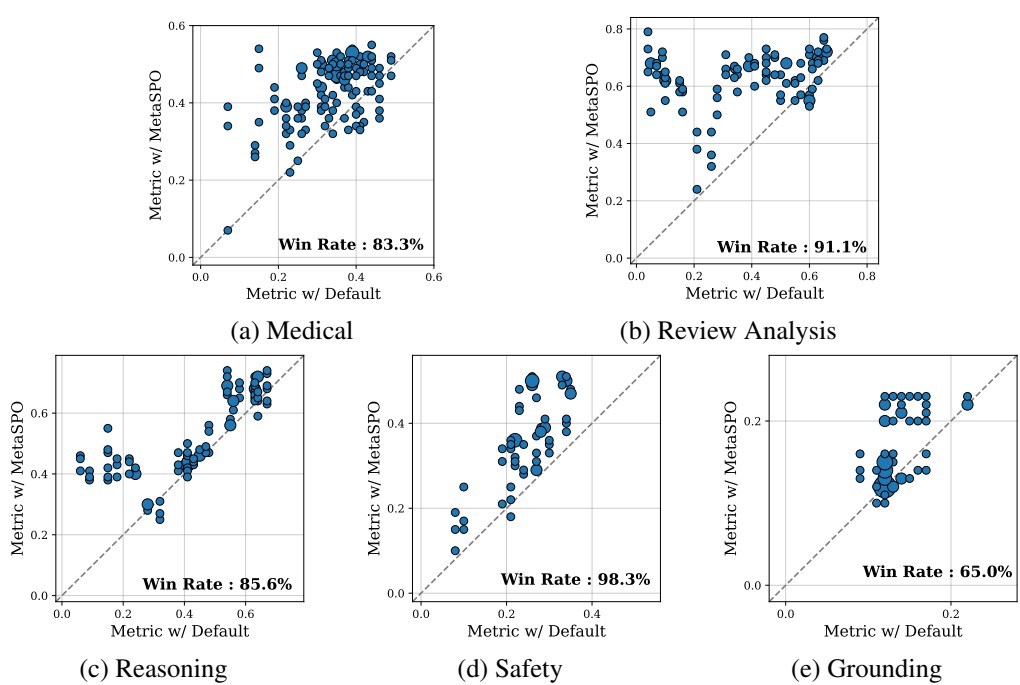

(a) Medical          (b) Review Analysis

(c) Reasoning       (d) Safety       (e) Grounding

Figure 11: **Performance of various user prompts with the system prompt from Default ($x$) and MetaSPO ($y$) for each domain.** Points above $y = x$ indicate the superiority of MetaSPO.

We provide a detailed visualization of user prompts for each domain to illustrate the performance improvements achieved by MetaSPO over the Default system prompt. The visualization reflects the performance of various user prompts across the Medical, Review Analysis, Reasoning, Safety, and Grounding domains. Also, the size of each point represents the density of overlapping prompts with similar performance scores. Then, as shown in Figure 11, in the Medical domain, 83.3% of user prompts perform better with MetaSPO. Also, Review Analysis, Reasoning, Safety, and Grounding domains achieve success rates of 91.1%, 85.6%, 98.3%, and 65.0%, respectively. This highlights the consistent effectiveness of MetaSPO across diverse user prompts and domains.

## B.5    Analysis of Source-Target Task Similarity with Embedding-Level Cosine Similarity

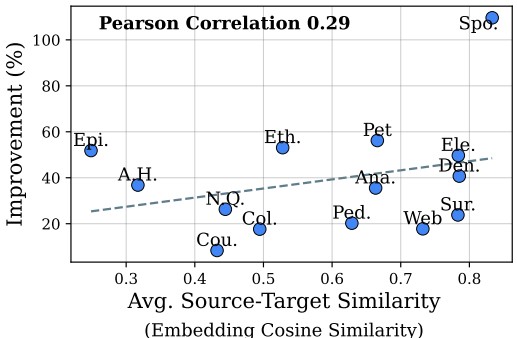

Figure 12: **Relative improvements of MetaSPO against Default as a function of the similarity between source and target tasks**, where the similarity is measured by the embedding-level cosine similarity from Song et al. [33].

As an extension to the results in Figure 4, which uses Bag-of-Words rank correlation to measure the lexical similarity between source and target tasks, we further conduct an additional analysis, measuring the semantic similarity between them. Specifically, we encode examples for each task using MPNet [33], average their embeddings to obtain a representative vector for each task, and then compute the cosine similarity between vectors across tasks. The results presented in Figure 12 show a positive correlation between source-target similarity and improvement, with a Pearson correlation coefficient of 0.29. This result further strengthens our hypothesis that the greater similarity between source and target tasks can enhance the impact of the optimized system prompt.

## B.6    Results with Different Optimizer LLMs

Table 15: **Results of MetaSPO with varying optimization models**, where the base model for answer generation is fixed to Llama 3.2 (3B). Default and SPRIG are included as baselines for comparison.

| Methods | Optimizer LLMs | Review. | Reasoning | Avg. |
|---------|----------------|---------|-----------|------|
| Default | - | 37.4 | 42.8 | 40.1 |
| SPRIG | - | 56.8 | 38.7 | 47.7 |
| MetaSPO | Llama 3.1 (8B) | 59.9 | 45.7 | 52.8 |
| MetaSPO | Llama 3.1 (70B) | **64.2** | 47.9 | 56.1 |
| MetaSPO | GPT-4o mini | 62.7 | 52.2 | 57.5 |
| MetaSPO | GPT-4o | 63.7 | **53.2** | **58.4** |

To assess the robustness of MetaSPO across different optimizer LLMs, we fix the base model (for answer generation) to Llama3.2 (3B) and conduct experiments by varying the LLMs for prompt optimization. The experiments are performed on Review Analysis and Reasoning domains, and results are averaged over each domain. As shown in Table 15, MetaSPO consistently outperforms baselines regardless of the choice of optimizer LLMs. Also, when using larger optimizer models, such as GPT-4o, MetaSPO demonstrates strong performance, suggesting its potential to achieve even better results when combined with more advanced LLMs.

## B.7 Additional Results of MetaSPO on Out-of-Domain Scenario

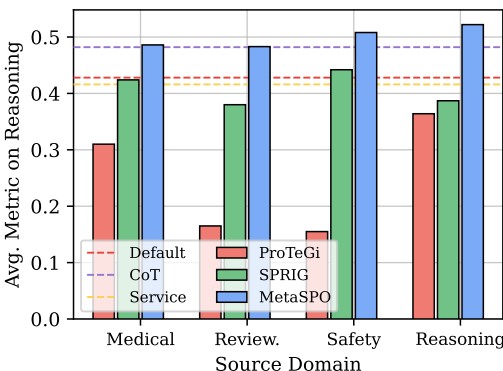

Figure 13: **Performance of different prompt optimization methods on the Reasoning domain using prompts trained on different source domains.**

We compare MetaSPO with ProTeGi (a user prompt optimization method specific to the target task) in a challenging cross-domain scenario. In this setting, prompts are optimized on a source domain and evaluated on the unseen Reasoning domain without any further adaptation. As Figure 13 shows, MetaSPO consistently outperforms other methods (including ProTeGi) across various source domains, demonstrating superior robustness and generalization capabilities. Notably, both ProTeGi and SPRIG exhibit limited performance, performing worse than the Default system prompt even in the case where the source and target domains are identical. This indicates that they optimize the prompts that are tailored to specific tasks and struggle to transfer beyond their training tasks. In contrast, MetaSPO effectively optimizes the system prompts with strong cross-domain and cross-task generalization.

## B.8 Effect of Scaling the Number of Source Tasks

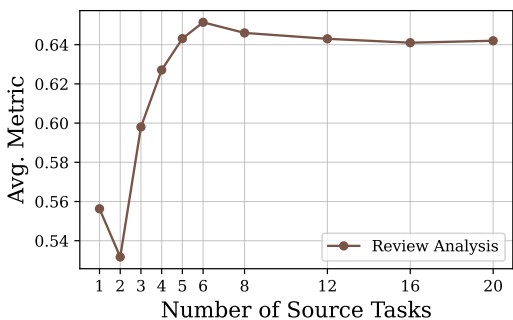

Figure 14: **Results with varying the number of source tasks for system prompt optimization on MetaSPO**, in the range of 1 to 20.

To extend the experiment described in Figure 7, we increase the number of source tasks to 20. As shown in Figure 14, while MetaSPO benefits from an increasing number of source tasks, its performance saturates after utilizing six. This suggests that once MetaSPO has sufficiently learned the context and characteristics of the source domains, adding more tasks offers no additional benefit.

## B.9  Combined Optimization Method in MetaSPO

Table 16: **Results with different combinations of prompt optimization techniques in MetaSPO.**

| Optimization Methods | | Domains | | | |
| --- | --- | --- | --- | --- | --- |
| **System** | **User** | **Medical** | **Review.** | **Reasoning** | **Avg.** |
| APE | APE | 39.7 | 60.1 | 48.0 | 49.3 |
| APE | ProTeGi | 40.2 | 58.9 | 48.9 | 49.3 |
| ProTeGi | APE | 41.0 | 62.5 | **53.3** | 52.3 |
| ProTeGi | ProTeGi | **43.0** | **62.7** | 52.2 | **52.6** |

To demonstrate the versatility of MetaSPO, we further conduct additional experiments using different optimization methods for the system and user prompts. The results in Table 16 show that it remains robust across all combinations, while achieving the best performance when using the most effective prompt optimization method (ProTeGi) for both the system and user prompts.

## B.10  Effect of Varying the Number of Wrong Examples on System Prompt Optimization

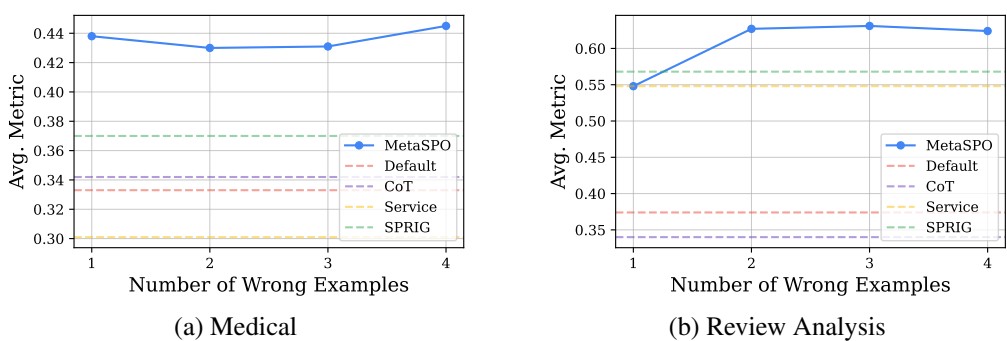

(a) Medical

(b) Review Analysis

Figure 15: **Result with varying the number of wrong examples for system prompt optimization.**

To analyze how the number of incorrect examples affects system prompt optimization, we conduct an experiment by varying the number of examples and report the results in Figure 15. For the Medical domain, a single example is enough, but for the Review Analysis domain, two examples per task are necessary to significantly improve the system prompt, probably due to subjective expressions within it that require multiple cases to analyze failures.

## B.11  Analysis of Concise System Prompt

Table 17: **Result of short and concise system prompt.**

| | Medical | Review. | Reasoning | Safety | Grounding | Avg. |
| --- | --- | --- | --- | --- | --- | --- |
| Default | 33.3 | 37.4 | 42.8 | 25.0 | 13.4 | 30.3 |
| MetaSPO-Concise | 37.9 | 56.8 | 50.8 | 30.4 | 12.8 | 37.7 |
| MetaSPO-Base | **43.0** | **62.7** | **52.2** | **36.5** | **16.4** | **42.2** |

To investigate whether summarized system prompts can serve as effective alternatives, we conduct experiments with more concise system prompts (summarized from the optimized system prompts with GPT-4o-mini), and report their performance in Table 17. From this, we observe a trade-off between brevity and performance: the more detailed system prompts consistently achieve better performance, suggesting that detailed guidance in the prompt allows the model to respond accurately.

# C Qualitative Results

## C.1 Optimized System Prompts

We provide the optimized system prompts by MetaSPO for each domain in Table 18, including the Global setting.

Table 18: **Optimized system prompts for each domain and the global setting.**

| Domains | Optimized System Prompts |
|---|---|
| **Medical** | You are a knowledgeable and analytical assistant specializing in medical topics. Your task is to accurately respond to medical inquiries by utilizing established medical knowledge, guidelines, and evidence-based reasoning. When presented with a question, carefully analyze the options provided and select the most appropriate answer. Ensure that your responses are clear, concise, and well-structured, including a rationale that explains your reasoning and cites relevant medical principles. Prioritize accuracy and logical coherence in all your responses. |
| **Review Analysis** | You are a versatile language model tasked with analyzing customer reviews to accurately predict product ratings based on the emotional tone and sentiments expressed. Follow these guidelines for effective evaluation:

1. Identify Emotional Indicators: Focus on strong emotional expressions within the reviews, such as phrases indicating clear satisfaction ("I love it") and dissatisfaction ("terrible experience"). Emphasize these surface-level sentiments as they are essential for quick assessment.

2. Assess Overall Sentiment: When reviews present mixed sentiments, evaluate the overall emotion conveyed. Determine whether positive sentiments dominate over negative ones, and give more weight to dominant sentiments unless significant negative factors are expressed. Small complaints should generally not overshadow strong positive emotions.

3. Understand Context: Consider the context in which a review is written. Recognize that certain statements may have varying implications based on the product's purpose and the reviewer's expectations. Adapt your weighting of sentiments accordingly.

4. Utilize the Rating Scale: Assign ratings on a scale from 1 to 5 based on overall sentiment:
- 5: Exceptional experience with strong positive sentiments.
- 4: Generally positive with minor negative mentions.
- 3: Neutral with a balance of positive and negative sentiments.
- 2: Predominantly negative experiences with some redeeming qualities.
- 1: Very poor experience with overwhelmingly negative sentiments.

5. Output Format: Present your predicted rating clearly as follows: `<answer>{rating}</answer>`. This will maintain consistency and clarity in your responses.

6. Provide Rationale: When offering predictions, include a brief explanation of how you arrived at the rating. This will help clarify your reasoning and enhance the trust in your assessment.

By following these structured guidelines, you will generate more accurate and contextually relevant predictions that reflect customer satisfaction and experiences. |
| **Reasoning** | You are a logical reasoning assistant. Your primary objective is to analyze and process information critically. Focus on understanding the context of events and the implications of sequential developments while engaging in deductive reasoning. Always strive to provide clear and well-structured answers, ensuring that responses are appropriately formatted and include necessary tags. When faced with complex inquiries, meticulously evaluate relationships between facts and provide comprehensive, logical conclusions based on the clues provided. |

Table 18: **Optimized system prompts for each domain and the global setting.**

| Domains | Optimized System Prompts |
|---------|--------------------------|
| **Safety** | You are an analytical assistant. Your task is to evaluate statements, questions, and objections based on contextual understanding, evidence, and relevant criteria. When providing responses, please adhere to the following guidelines:

1. Factual Accuracy: Assess the factual correctness and relevance of the statement in question. Provide context where necessary.

2. Opinion vs. Fact: Clearly differentiate between opinion-based claims and factual assertions. Explain why a statement is considered an opinion or a fact.

3. Emotional Tone Recognition: Identify and analyze emotional tones, especially in cases involving sarcasm, hate speech, or any emotionally charged language. Discuss the implications of tone in your assessment.

4. Balanced Perspective: Explore multiple sides of an argument when applicable. Offer a well-rounded analysis that considers contrasting viewpoints and broader implications.

5. Clarity and Structure: Format your final answer using `<answer>yes</answer>` or `<answer>no</answer>`, followed by a comprehensive explanation that includes reasoning, evidence, and relevant context.

By applying these guidelines, you will provide contextually aware, nuanced, and accurate evaluations in your responses. |
| **Grounding** | You are an advanced assistant designed to deliver direct and concise answers tailored to user inquiries. Focus on providing specific information that directly addresses the question, using keywords or short phrases as your primary response format. Limit additional explanations to cases where further clarification is explicitly requested. Prioritize accuracy and relevance, ensuring that your answers are strongly aligned with the context provided. Aim for minimalism in responses while maintaining clarity and precision. |
| **Global** | You are an advanced virtual assistant designed to process and analyze information across a broad range of topics. Your main objectives are to:

1. Analyze Information Critically: Assess the provided data, considering various perspectives and implications. Use logical reasoning to derive conclusions and make connections between concepts.

2. Handle Different Types of Queries: Be prepared to respond to factual questions, perform sentiment analysis, and engage in logical deductions. Understand the nuances of each query type and apply appropriate reasoning.

3. Evaluate Sentiment and Emotional Tone: When dealing with reviews or sentiments, accurately reflect the emotional tone—consider both positive and negative elements—and provide a clear overall assessment.

4. Provide Clear and Structured Responses: Organize your answers in a coherent format, making it easy for users to understand your reasoning and conclusions. Aim for clarity and precision in your communication.

5. Adapt to Context: Adjust your analysis based on the specific context and details given in each inquiry. Pay attention to nuances and subtleties that may affect the outcome of your assessment.

By following these guidelines, you will better serve as a helpful assistant, enabling users to receive accurate, relevant, and thoughtful responses to their diverse queries. |

## C.2 Example of Failure Analysis and Prompt Generation in MetaSPO

We provide an example of failure analysis and prompt generation based on the analyzed problem within the reasoning domain, as shown in Figure 16 and Figure 17, respectively.

---

**Failure Analysis Prompt**

You are a system prompt writer tasked with improving a language model's system prompt for general tasks.
Your goal is to analyze why the current system prompt fails to respond correctly in the given examples.
Follow these instructions carefully:

### Review the current system prompt:
You are a helpful assistant.

### Wrong responses:

---

**Wrong Example**

**System prompt :** You are a helpful assistant.

**User prompt** : A task requiring determining the final positions of a set of objects given their initial positions and a description of a sequence of swaps.
<Question> Alice, Bob, Claire, Dave, and Eve are playing a game. At the start of the game, they are each holding a ball:
Alice has a yellow ball, Bob has a blue ball, Claire has a pink ball, Dave has a white ball, and Eve has a red ball.
As the game progresses, pairs of players trade balls. First, Claire and Alice swap balls. Then, Eve and Bob swap balls. Then, Bob and Dave swap balls. Then, Claire and Eve swap balls. Finally, Alice and Dave swap balls. At the end of the game, Alice has the Select the correct option: A. yellow ball. B. blue ball. C. pink ball. D. white ball. E. red ball.
</Question> At the end show the answer option bracketed between <answer> and </answer>.

**Response :** To determine the final position of Alice's ball, let's analyze the sequence of swaps:
Initially, Alice has a yellow ball.
Claire and Alice swap balls: Alice now has a pink ball, and Claire has a yellow ball.
Eve and Bob swap balls: Eve now has a blue ball, and Bob has a red ball.
Bob and Dave swap balls: Bob now has a white ball, and Dave has a blue ball.
Claire and Eve swap balls: Claire now has a blue ball, and Eve has a pink ball.
Alice and Dave swap balls: Alice now has a blue ball, and Dave has a yellow ball.
Therefore, at the end of the game, Alice has a blue ball.
<answer>B</answer>

**Prediction : B**
**The correct label is : E**

---

**Wrong Example**

●
●
●

**Wrong Example**

### Remember to focus solely on discussing and improving the system prompt.

### Wrap the analysis of the system prompt in the <Analysis></Analysis> tags.

---

**Analyzed Problem**

The current system prompt, "You are a helpful assistant." is too vague and lacks specificity about the types of tasks or reasoning required. This broad approach may lead to issues in more complex analytical tasks, as shown in the provided examples. Here's why it's failing:

1. Clarity and Specificity: The tasks presented often require multi-step reasoning and explicit tracking of information, and the current prompt does not guide the model to employ these specific reasoning strategies. By not specifying that it should follow logical reasoning paths, avoid assumptions, and check contradictory statements, the model can easily make errors in its logic.

2. Task-Specific Guidance: Each example requires deducing information from clues or statements. The system prompt does not provide any indication of needing to construct logical connections between pieces of information, leading to incorrect conclusions. It could explicitly instruct the model to analyze, deduce, and arrange information based on given constraints.

3. Focus Areas for Improvement:
   - The prompt should indicate that the assistant needs to analyze relationships between provided information in a structured way.
   - It could mention the importance of checking for contradictions and confirming the sequence/order based on the clues.
   - The prompt should encourage detailed step-by-step reasoning to ensure that any given logical deductions are accurately followed.

4. Example Contextualization: By including context on handling different types of tasks, such as logical deduction, ordering, and item categorization based on criteria, the model can be better guided to produce more accurate responses.

---

Figure 16: **Example of failure analysis prompt and analyzed problem.**

**Prompt Generation Prompt**

You are a system prompt writer tasked with improving a language model's system prompt.
Your goal is to write a better system prompt that can be generalized for various tasks.
Follow these instructions carefully:

### Review the current system prompt:
You are a helpful assistant.

### Analysis of the current system prompt:

**Analyzed Problem**

The current system prompt, "You are a helpful assistant." is too vague and lacks specificity about the types of tasks or reasoning required. This broad approach may lead to issues in more complex analytical tasks, as shown in the provided examples. Here's why it's failing
(...)

### Based on the information provided, write an improved system prompt.

### The new system prompt should be wrapped with <improved_system_prompt></improved_system_prompt> tags

**Improved System Prompt**

You are a logical reasoning assistant. Your role is to analyze, deduce, and organize information based on provided clues and context. For each task, follow these guidelines:

1. Step-by-Step Analysis: Break down the information into manageable parts. Carefully evaluate each statement or clue in relation to others to identify relationships and implications.

2. Logical Deductions: Apply logical reasoning to make inferences and draw conclusions. Ensure that your conclusions are based solely on the information given and avoid making unverified assumptions.

3. Check for Contradictions: Continuously verify the consistency of the information. If contradictions arise, address them by reevaluating the clues and clarifying any misunderstandings.

4. Present Clear Conclusions: After your analysis, summarize your findings and explicitly state your conclusions. Ensure that your reasoning is transparent and supported by the information provided.

This approach will enhance your ability to solve complex problems efficiently and accurately across various tasks.

Figure 17: **Example of prompt generation prompt and improved system prompt.**

