# OpenReview forum: "System Prompt Optimization with Meta-Learning"
_NeurIPS.cc/2025/Conference — NeurIPS 2025 poster_

### Official Review · Reviewer_tfVR · 2025-07-01

**Clarity:** 2
**Significance:** 2
**Originality:** 2
**Rating:** 4
**Confidence:** 4

**Summary:**

This paper proposes MetaSPO, a bilevel prompt optimization framework designed to optimize system prompts for large language models (LLMs), which are often hand-crafted and static. Unlike prior work that focuses on optimizing user prompts for specific tasks, MetaSPO
separates prompt optimization into two levels: an inner loop that updates user prompts based on task-specific failure cases, and an outer loop that updates the system prompt based on aggregated failure patterns across tasks. The framework leverages the LLMs to analyze
incorrect outputs and generate improved prompt candidates. Experiments on 14 unseen datasets across 5 domains demonstrate improved generalization and reduced adaptation cost compared to baseline prompting approaches.

**Questions:**

- How is response correctness evaluated during prompt selection? Are labels used?
- Are the top-k system prompts ultimately ensembled, averaged, or is a single prompt selected?
- Why was only LLaMA 3.2B used as the generator? Would larger models reduce reliance on the analyzer?
- In experiments where in-domain prompts underperform global prompts, what explains this result?

**Ethical Concerns:**

["NO or VERY MINOR ethics concerns only"]

**Final Justification:**

After reading the authors' response, I am happy to raise the score to borderline accept.

**Limitations:**

Yes, in the appendix, there is a discussion on limitations. However, no direction is presented to address the limitation.

**Quality:**

3

**Strengths And Weaknesses:**

Strength:
- Proposes an interesting idea and well-motivated problem: system prompt optimization using a meta-learning framework
- Demonstrates cross-domain generalization on 14 unseen datasets.

Weakness:
- The idea of using LLMs to optimize prompts is not fresh. Although the paper introduces system prompt optimization techniques,  the approach resembles components used in prior works [1]. More explanation, comparison, and contrast are required to highlight the novelty of the approach.
- The meta-learning claim is overstated: The method is more accurately a search-based approach using LLM for optimization than the classic meta-learning algorithm, such as MAML. More explanation and evidence are required to justify the claim
- Lack of clarity on implementation details. For example, regarding top-k prompt usage, it is unclear whether the top-k system prompts
are ensembled, re-ranked, or reduced to a single selection in the final evaluation. Terms like “analyzer” and “optimizer” are used
interchangeably but are not clearly defined. The paper seems not to explain how response correctness is measured during optimization (e.g., automatic metrics, reward model, or human evaluation?).
- The practicality of the approach is questionable. It highly depends on strong external analyzers (e.g., GPT-4), which may limit accessibility. The generator's limited capacity may overstate the impact of the optimized system prompt. Also, the computational cost of repeated prompt generation and evaluation could be significant, given that there is only a modest performance gain.
- The baseline set seems to ignore the recent user prompt optimization approaches [2,3,4]. To prove that a complex bilievel prompt optimization is needed, there should be a significant gain observed compared to these baselines.

[1] Agarwal, Eshaan, Joykirat Singh, Vivek Dani, Raghav Magazine, Tanuja Ganu, and Akshay Nambi. "PromptWizard: Task-Aware Prompt Optimization Framework." arXiv preprint arXiv:2405.18369 (2024).

[2] Yongchao Zhou, Andrei Ioan Muresanu, Ziwen Han, Keiran Paster, Silviu Pitis, Harris Chan, and Jimmy Ba. Large language models are human-level prompt engineers. In The Eleventh International Conference on Learning Representations, 2023.

[3] Zhang, Tianjun, Xuezhi Wang, Denny Zhou, Dale Schuurmans, and Joseph E. Gonzalez. "TEMPERA: Test-Time Prompt Editing via Reinforcement Learning." In The Eleventh International Conference on Learning Representations, 2023.

[4] Do, VT. et al. (2025). Automatic Prompt Selection for Large Language Models. In: Wu, X., et al. Advances in Knowledge Discovery and Data Mining. PAKDD 2025. Lecture Notes in Computer Science(), vol 15872.

---

> ### Author Rebuttal · Authors · 2025-07-31
>
> We sincerely appreciate your constructive and helpful comments, as well as your acknowledgement that we propose a well-motivated problem of system prompt optimization using meta-learning, and we demonstrate cross-domain generalization on 14 unseen datasets. We have made every effort to address all your comments faithfully.
>
> ---
>
> > W1. Using LLMs to optimize prompts is not fresh. Although the paper introduces system prompt optimization techniques, the approach resembles components used in prior work [1].
>
> Thank you for raising the concern about novelty and for pointing out the relevant work. While we agree that prompt optimization is a well-studied area, our work introduces a novel problem: bilevel system prompt optimization, which aims to learn a single, task-agnostic system prompt that generalizes across diverse user prompts and unseen tasks. This setting is fundamentally different from prior works [1], which require labeled data and re-optimization for each task, ultimately optimizing user prompts specific to each task. Also, in MetaSPO, even without targeted data on diverse tasks and domains, a single global-level system prompt obtained is generalizable to them, which we validate in Table 1 and Figure 5.
>
> ---
>
> > W2. The meta-learning claim is overstated, since the method is more like a search-based approach using LLM for optimization than classic meta-learning (e.g., MAML).
>
> We would like to clarify that meta-learning is not limited to classic gradient-based methods like MAML; rather, it broadly refers to learning shared knowledge from a distribution of tasks to enable generalization to unseen tasks. In this sense, MetaSPO fits well within the meta-learning paradigm: it employs bilevel optimization where the outer loop optimizes a task-agnostic system prompt that captures transferable knowledge across tasks, along with task-specific user prompt optimization for each task in the inner loop. In addition, this learned task-agnostic system prompt generalizes to unseen tasks and enables faster adaptation at test time, which is the core objective and key characteristic observed in meta-learning.
>
> ---
>
> > W3.1. Regarding top-k prompt usage, it is unclear whether the top-k system prompts are ensembled, re-ranked, or reduced to a single selection in the final evaluation.
>
> We select and use the single best-performing system prompt based on its average performance across tasks and user prompts, after generating multiple candidate system prompts through the outer loop (See Algorithm 3 and Line 14 in the supplementary material).
>
> > W3.2. Analyzer and Optimizer are used interchangeably, but are not clearly defined.
>
> In MetaSPO, the role of Analyzer is to identify failure patterns of prompts on incorrectly predicted samples (called problem analysis), and Generator is designed to produce improved prompt candidates from the failure analysis (as shown in Algorithm A.3 and Tables 6, 7, 8, and 9). Both are implemented by instantiating the LLM with their corresponding prompt templates. In addition, the term Optimizer (in the caption of Table 4) refers to the specific prompt optimization techniques (such as APE or Evo), and MetaSPO is agnostic to the choice of them. We will make sure to improve their descriptions in the revision.
>
> > W3.3. How is response correctness measured during optimization?
>
> We measure the response correctness using task-specific automatic metrics, such as Exact Match (EM) and F1, depending on the datasets (See Appendix A.1). Also, the answers are extracted from the model outputs with the predefined answer format, such as &lt;answer&gt;{answer}&lt;/answer&gt;.
>
> ---
>
> > W4.1.The approach depends on strong external analyzers, which may limit accessibility.
>
> The analyzer can also be instantiated with less powerful language models, such as LLaMA 3.1 (8B) or GPT-4o mini, as demonstrated in our experimental results in Table 15. Specifically, MetaSPO consistently shows substantial gains over baselines, regardless of the choice of language models for operationalizing the analyzer, which suggests that it is indeed accessible and practical for a wide range of users with various computational budgets.
>
>
> > W4.2. The generator's limited capacity may overstate the impact of the optimized system prompt.
>
> We would like to point out that our experiments cover a wide range of generators, including Llama (8B), Qwen 2.5 (7B), and GPT-4o mini, in addition to Llama (3B), as shown in Table 3. From this, we then show that MetaSPO consistently outperforms baselines with varying generator models, which confirms its generalizability and robustness. However, to more fully address your comment, we further conduct experiments with GPT-4o, evaluating the system prompt optimized from GPT-4o mini to GPT-4o without further optimization, and as shown in Table D.1 below, MetaSPO achieves the highest average score over baselines, indicating its strong transferability to even the strongest model. Lastly, we believe that the ability of MetaSPO in improving the performance of lightweight LLMs is indeed its strength, which offers practical values in terms of efficiency, especially in resource-constrained settings.
>
> Table D.1. Unseen generalization results of GPT-4o with system prompts optimized from GPT-4o mini
> ||Review|Analysis|||Reasoning|||
> |:---:|:---:|:---:|:---:|:---:|:---:|:---:|:---:|
> ||Electronics|Pet|Sports|Count|Epistemic|Color|Avg.|
> |Default|73.4|75.0|69.2|78.2|81.6|98.0|79.2|
> |CoT|73.0|76.0|70.6|80.0|80.0|98.4|79.7|
> |Service|70.6|75.0|66.8|82.0|81.6|98.0|79.0|
> |SPRIG|73.0|73.8|68.0|81.8|81.8|98.0|79.4|
> |MetaSPO|76.9|75.4|77.3|82.9|81.1|98.9|**82.1**|
>
>
> > W4.3. The computational cost of repeated prompt generation and evaluation could be significant, given that there is only a modest performance gain.
>
> We appreciate your comment. However, we first would like to clarify that the process of system prompt optimization can be done only once, and the resulting system prompt can be reusable across diverse tasks and domains with high efficiency for adapting to them, which is similar to how modern foundation models are pre-trained once and then fine-tuned or prompted for various downstream applications. Therefore, considering the total computation cost (system and user prompt optimizations) might be less relevant in practical settings.
>
> Nevertheless, we further show in Table D.2 that MetaSPO becomes more efficient than ProTeGi as the number of tasks increases. Specifically, although MetaSPO incurs a higher initial cost for optimizing the system prompt on diverse source tasks, the amortized cost of the system prompt becomes increasingly negligible: with just 3 target tasks, it requires fewer total model calls per task (for both base and optimizer models) than ProTeGi.
>
> Lastly, the performance gain from MetaSPO is not modest but statistically significant on 11 out of 14 tasks (Table 13), which improves user prompts in 85% of cases (Figure 3).
>
> Table D.2. Comparison of computational cost per target task between ProTeGi and MetaSPO with ProTeGi.
> | # of Target Tasks | Methods | # of Optimizer Model Calls | # of Base Model Calls |
> |:---:|:---:|:---:|:---:|
> |1|ProTeGi(20 iterations)|366|9150|
> |0|MetaSPO(Pre-train on 4 tasks)|126|18000|
> |1|MetaSPO + ProTeGi(4 iters)|**204**|19950|
> |2|MetaSPO + ProTeGi(4 iters)|141|10950|
> |3|MetaSPO + ProTeGi(4 iters)|120|**7950**|
>
> ---
>
> > W5. The baselines seem to ignore the recent user prompt optimization approaches [2-4].
>
> We would like to emphasize that MetaSPO targets task-agnostic system prompt optimization, unlike the suggested task-specific user prompt optimization approaches; therefore, they should be considered as orthogonal and indeed complementary to our framework, as it is compatible with them in optimizing prompts (as shown in Table 4).
>
> Moreover, we conduct test-time adaptation experiments to validate that MetaSPO is indeed helpful to further improve the performance when coupled with the user prompt optimization techniques. The results in Table D.1 show notable gains in performance when using APE [2] and PromptAgent [5] for test-time adaptation, supporting the orthogonal and complementary effect of our MetaSPO, which even query-dependent prompt optimization approaches [3-4] could similarly benefit from: MetaSPO offers consistent, task-agnostic guidance that improves stability and reduces variance across tasks and user prompts. We will discuss it.
>
> Table D.3. Test-Time Adaptation with other user prompt optimization methods
> |Optimization Strategy|System Prompt|Medical|Review.|Reasoning|Safety|Grounding|Avg.|
> |:---:|:---:|:---:|:---:|:---:|:---:|:---:|:---:|
> |APE|Default|0.455|0.643|0.609|0.395|0.168|0.479|
> ||MetaSPO|**0.471**|**0.680**|**0.640**|**0.437**|**0.175**|**0.505**|
> |PromptAgent|Default|0.451|0.654|0.614|0.532|0.175|0.502|
> ||MetaSPO|**0.465**|**0.685**|**0.663**|**0.593**|**0.183**|**0.533**|
> |APO|Default|0.451|0.689|0.640|0.599|0.175|0.524|
> ||MetaSPO|**0.456**|**0.714**|**0.673**|**0.672**|**0.199**|**0.552**|
>
> [5] PromptAgent: Strategic Planning with Language Models Enables Expert-level Prompt Optimization, ICLR 2024
>
> ---
>
> > Q1. How is response correctness evaluated?
>
> Please refer to our response to W3.3.
>
> ---
>
> > Q2. How are the top-k system prompts ultimately used?
>
> Please refer to our response to W3.1.
>
> ---
>
> > Q3. Why was only LLaMA 3.2B used as the generator, and how about larger models?
>
> Please refer to our response to W4.2.
>
> ---
>
> > Q4. What explains this result: in-domain prompts underperform global prompts.
>
> This is because the global prompts benefit not only from exposure to in-domain tasks but also from additional knowledge and reasoning patterns acquired from diverse out-of-domain tasks.
>
> ---
>
> > L1. While there is a discussion on limitations, no direction is presented to address them.
>
> In our discussion of limitations, we also outline a potential direction to address them: distilling stronger optimizer LLMs into smaller ones to improve the capability of lightweight optimizers.

---

> > ### Comment · Reviewer_tfVR · 2025-08-07
> > **Response to the rebuttal**
> >
> > Thank you for your detailed rebuttal. I will consider your answers to revise the rating accordingly.

---

> > > ### Author Response · Authors · 2025-08-07
> > >
> > > Thank you very much for taking the time to review our paper and for considering our rebuttal in your evaluation.
> > >
> > > With that in mind, we have made our best effort to address your comments and concerns in great detail in our responses, and wanted to make sure that everything you raised has been sufficiently clarified. If there are any parts that you feel remain unclear or insufficiently addressed, please do not hesitate to let us know; we would be more than happy to further clarify. Otherwise, in line with your kind note that you would consider revising the rating accordingly, we would greatly appreciate it if you could update your score.
> > >
> > > We thank you once again for your time and effort, as well as your constructive feedback, which has been invaluable in shaping our work, and we will make sure to incorporate your suggestions into the final version.

---

> > > ### Author Response · Authors · 2025-08-08
> > >
> > > Dear Reviewer tfVR,
> > >
> > > We remain deeply appreciative of your review and insightful feedback. Following up on your earlier note that you would consider revising your rating based on our rebuttal, we wanted to gently check whether there are any remaining points you would like us to clarify. If there are, please let us know, and if not, we would be truly grateful if you could spare just a moment to update your evaluation accordingly. Thank you again for your time and consideration.
> > >
> > > Warm regards, Authors

---

### Official Review · Reviewer_rqST · 2025-07-01

**Clarity:** 4
**Significance:** 3
**Originality:** 2
**Rating:** 5
**Confidence:** 4

**Summary:**

This paper takes the concept of task-based prompt optimization, and extends it to include optimization of the system prompt as well as of the task-specific user prompt. By performing these optimizations together, and across a wide variety of tasks, this system creates a system prompt that helps across a variety of task types, including unseen tasks, and allows for faster optimizing of task-specific user prompts with increased performance.

**Questions:**

How was it decided how many system prompts and user prompts would be generated and maintained in each loop iteration?

How does the total processing required compare to existing task-specific techniques (not just in terms of adding a new task once you have a system prompt, but in terms of training the system from scratch for a variety of tasks)?

**Ethical Concerns:**

["NO or VERY MINOR ethics concerns only"]

**Final Justification:**

Keeping my score as Accept. Although not a theoretical breakthrough or out-of-the-box idea, this work is important in showing that system prompts can/should be part of the optimization process, and that doing this helps efficiency in task learning as the number of tasks increases.

The responses to the comments by all reviewers were thorough and helpful. Clarifying the efficiency of the method including both prompt optimizations (although one is held steady after learning even as new tasks are added) and explaining more clearly why the loops use the number of candidate prompts helps fill out the paper information.

**Limitations:**

yes

**Quality:**

3

**Strengths And Weaknesses:**

This paper is very clearly written and the technique described is explained well. It clearly spells out 2 separate benefits: the ability to help with unseen tasks, and faster optimization of task-specific prompts with excellent performance compared to task-only (no system prompt) optimization. The work is well-motivated and the technique is clearly explained. The experiments done and their results clearly show the benefit of this meta-optimization, especially in terms of being effective across new/unseen/unrelated tasks. Analyzing the attention to system prompt vs user prompt helps the reader understand the value in optimizing system prompts. It was helpful to include the analysis of using a prompt optimized with one model on another. One could imagine future work trying to optimize a global system prompt that is model-agnostic by using a technique like this on multiple models at the same time within in the inner and outer loops.

The idea itself isn't incredibly novel (basically extending existing prompt optimization methods to consider the system prompt in an outer loop), but I haven't seen it done in this way before, and it is clear from the results that it has value.

One thing missing in this paper is an analysis of how much processing this technique requires versus existing task-only techniques. It is clear from the paper that training a prompt for a new task, given a trained system prompt, is very efficient and this is backed up by some numbers. But, in terms of total processing to accomplish some number of tasks, how does the total processing compare in terms of having to optimize both the system prompt and the task prompts, versus using existing techniques to optimize only a task prompt for each task?

It is also not clear in the paper why the authors generate 9 different system prompts (maintaining one) and 3 different task prompts (maintaining them all) in the implementation details. It would be helpful to understand how those numbers were arrived at. Was there an empirical reason to choose them, or was it from tying different values and comparing results, or based on resource usage, or something else?

---

> ### Author Rebuttal · Authors · 2025-07-31
>
> We sincerely thank you for your valuable and constructive feedback. We are also grateful for your recognition of the strengths of our work: the work is well-motivated, the technique is clearly explained, and the experiments and their results clearly show the benefit of the meta-optimization, demonstrating the effectiveness of our approach in helping with unseen tasks and optimizing task-specific prompts rapidly with excellent performance. We have carefully considered and faithfully responded to all your comments.
>
> ---
>
> > W1. The idea itself is not incredibly novel (basically extending existing prompt optimization methods to consider the system prompt in an outer loop), but I have not seen it done in this way before, and it is clear from the results that it has value.
>
> Thank you for acknowledging the value of our work and its empirical effectiveness. While we agree that our work builds upon prior prompt optimization methods, our core novelty lies in introducing the bilevel system prompt optimization problem (which has not been previously formalized), and proposing a meta-learning framework designed to optimize system prompts in synergy with user prompts across diverse tasks, which we believe are valuable to the community since optimization of system prompts (despite their wide use) are underexplored.
>
> ---
>
> > W2. One thing missing is an analysis of how much processing this technique requires versus existing task-only techniques. While it is clear that training a prompt for a new task given a trained system prompt is very efficient, in terms of total processing to accomplish some number of tasks, how does the total processing cost in both system and user prompt optimization compare to existing techniques to optimize only a task prompt for each task?
>
> This is an excellent question. We first would like to clarify that the process of system prompt optimization can be done only once, and the resulting system prompt can be reusable across diverse tasks and domains with high efficiency for adapting to them, which is similar to how modern foundation models are pre-trained once and then fine-tuned or prompted for various downstream applications. Therefore, considering the total computation cost (system and user prompt optimizations) might be somewhat less relevant in practical settings.
>
> Nevertheless, in addition to the computational cost comparison provided in Appendix B.3 (where the number of model calls to optimization and generation models is used as a proxy for processing cost), we further show in Table C.1 that MetaSPO becomes more efficient than ProTeGi as the number of tasks increases. Specifically, although MetaSPO incurs a higher initial cost for optimizing the system prompt on diverse source tasks, the amortized cost of the system prompt becomes increasingly negligible: with just 3 target tasks, it requires fewer total model calls per task (for both base and optimizer models) than ProTeGi.
>
> Table C.1. Comparison of computational cost per target task between ProTeGi and MetaSPO with ProTeGi.
> | # of Target Tasks | Methods | # of Optimizer Model Calls | # of Base Model Calls |
> |:---:|:---:|:---:|:---:|
> | 1 | ProTeGi(20 iterations) | 366 | 9150 |
> | 0 | MetaSPO(Pre-train on 4 tasks) | 126 | 18000 |
> | 1 | MetaSPO + ProTeGi(4 iters) | **204** | 19950 |
> | 2 | MetaSPO + ProTeGi(4 iters) | 141 | 10950 |
> | 3 | MetaSPO + ProTeGi(4 iters) | 120 | **7950** |
>
> ---
>
> > W3. It is also not clear why the authors generate 9 different system prompts (maintaining one) and 3 different task prompts (maintaining them all) in the implementation details. It would be helpful to understand how those numbers were arrived at.
>
> We thank you for pointing it out and apologize for not providing a rationale for the implementation details about the number of prompts generated and maintained. In the inner loop for task prompt optimization, we generate 3 candidate user prompts per task, following the empirically validated, conventionally used number [1], and then retain the top 3 among the previous and current candidate prompts to ensure diversity in user prompts (which system prompts are jointly optimized with). In contrast, in the outer loop, we generate 9 candidate system prompts (i.e., 3 candidate prompts for each user prompt optimized from the prior step, using the conventional number), and retain only the top-performing one. We would like to note that this number is adopted to control the exponential growth in the number of prompts across iterations: if we were to retain two system prompts instead, the number of user prompts in the next inner loop would grow to 6 (2×3), and subsequently, 18 system prompts in the next outer loop (6×3). In other words, while generating and retaining more prompts could potentially improve performance, our design aims to strike a balance between effectiveness and computational efficiency, and exploring the impact of more aggressive prompt expansion with costs would be an interesting direction for future work.
>
> [1] PromptAgent: Strategic Planning with Language Models Enables Expert-level Prompt Optimization, ICLR 2024
>
> ---
>
> > Q1. How was it decided how many system prompts and user prompts would be generated and maintained in each loop iteration?
>
> We thank you again for your constructive comment. Please refer to our response to W3.
>
> ---
>
> > Q2. How does the total processing required compare to existing task-specific techniques?
>
> We thank you again for your valuable comment. Please refer to our response to W2.

---

> > ### Comment · Reviewer_rqST · 2025-08-05
> >
> > Thank you for your responses. The explanation of the numbers of prompts used/kept in each part of the algorithm is helpful. It is clarifying to state that the outer loop only keeps one candidate prompt to avoid exponential growth. Making this more clear in the paper would be helpful.
> >
> > I am keeping my score as Accept.

---

> > > ### Author Response · Authors · 2025-08-06
> > >
> > > Thank you for your thoughtful feedback and continued support in recommending the acceptance of our paper. We also appreciate your acknowledgment of our clarifications regarding the number of prompts selected, and we will revise the paper to more clearly explain this point as suggested.

---

### Official Review · Reviewer_bNQ9 · 2025-07-02

**Clarity:** 3
**Significance:** 2
**Originality:** 2
**Rating:** 3
**Confidence:** 4

**Summary:**

This paper presents a bilevel learning framework for optimizing the system prompt to enhance LLM performance. The authors decompose the input into three components: a task-agnostic system prompt, a task-specific user prompt, and a specific query. They optimize the system prompt across diverse user prompts and tasks, while also iteratively refining user prompts—using the optimized user prompts to further improve the system prompt. The proposed method is evaluated through extensive experiments on 14 unseen datasets spanning 5 different tasks.

**Questions:**

1. Do the authors conduct any case studies to illustrate how the model's output changes from incorrect to correct? For example, is the improvement mainly driven by format-related instructions in the optimized system prompt, as suggested by its inclusion of output format in Appendix?
2. Have the authors tried applying their method to more capable models (e.g., GPT-4o-latest or o3) or more challenging tasks (e.g., AIME 2024 or LiveBench)?
3. As mentioned in Line 240, the authors use ten user prompts per task. Could the authors provide detailed results using only these ten user prompts (e.g., in the form of a table similar to Table 1) to isolate the impact of system prompt optimization, beyond what is shown in Figure 3?
4. In Figure 6, why does the performance of the default system prompt vary across different iterations and example counts?

**Ethical Concerns:**

["NO or VERY MINOR ethics concerns only"]

**Limitations:**

Yes.

**Paper Formatting Concerns:**

No.

**Quality:**

2

**Strengths And Weaknesses:**

Strengths
1. The paper is motivated by a reasonable and important premise: optimizing the often-overlooked system prompt to improve LLM performance.
2. The experiments are comprehensive, covering a variety of evaluation settings with detailed analyses supporting the proposed method’s efficacy.

Weaknesses
1. The paper lacks theoretical insights into why meta-learning is effective for system prompt optimization and how it enables performance improvements.
2. The tasks included are limited to deterministic problems. The framework’s effectiveness on subjective tasks—such as creative writing and open-ended dialogue—remains unclear, which is notable given that system prompts are used in such chat scenarios (e.g., ChatGPT).

---

> ### Author Rebuttal · Authors · 2025-07-31
>
> We sincerely thank you for your constructive and helpful comments, as well as your recognition of the strengths of our work, including our motivation to optimize overlooked system prompts and our comprehensive experiments, supporting the efficacy of our method. We have carefully considered and faithfully responded to your comments.
>
> ---
>
> > W1. The paper lacks theoretical insights into why meta-learning is effective for system prompt optimization and how it enables performance improvements.
>
> We appreciate your feedback regarding the desire for a deeper theoretical understanding. However, as you acknowledged as strengths of our work, we provide comprehensive experiment results with detailed analyses, which support all the claims that we make: the system prompt learned from meta-learning improves generalization and accelerates adaptation. Specifically, we show consistent performance improvements across 14 unseen tasks in 5 domains, including cross-domain and cross-model generalization (Table 3 and Figure 5), demonstrating that meta-learning enables generalization over diverse domains and models. Also, the optimized system prompt via meta-learning enables the rapid and scalable adaptation, requiring fewer iterations and data at test time (Figure 6). Therefore, while a formal theoretical analysis is indeed an interesting direction for future work, we strongly believe that the lack of such insight should not be viewed as a limitation, given that our empirical evidence sufficiently provides concrete support for the efficacy of our method.
>
> ---
>
> > W2. The tasks included are limited to deterministic problems; thus, the effectiveness of the framework on subjective tasks remains unclear (while system prompts are also used in such chat scenarios).
>
> Thank you for raising this concern, and to address this, we evaluate the efficacy of the global system prompt optimized by MetaSPO on the MT-Bench dataset [1], which is designed for multi-turn, open-ended subjective tasks, such as Writing, Humanities, and Roleplay. Then, as shown in Table B.1 below, we observe that MetaSPO consistently outperforms baselines (including SPRIG and CoT) across tasks, with GPT-4o mini for evaluation. In other words, these findings provide strong empirical evidence that MetaSPO can apply to a diverse range of tasks, including deterministic and subjective tasks.
>
> Table B.1. Result of MetaSPO on MT-Bench [1]
> |Base Model|Method|Humanities|Roleplay|Writing|Avg.|
> |---|---|:---:|:---:|:---:|:---:|
> |Llama3.2 (3B)|Default|7.65|7.60|7.55|7.60|
> ||CoT|**7.70**|7.60|7.50|7.60|
> ||Service|7.65|6.75|7.60|7.33|
> ||SPRIG|7.60|7.30|7.60|7.50|
> ||MetaSPO|**7.70**|**7.65**|**7.85**|**7.73**|
> |Llama3.1 (8B)|Default|8.20|7.50|**7.75**|7.82|
> ||CoT|8.20|7.90|7.70|7.93|
> ||Service|7.80|7.75|7.65|7.73|
> ||SPRIG|**8.25**|7.85|7.60|7.90|
> ||MetaSPO|8.20|**8.05**|**7.75**|**8.00**|
>
> [1] Judging LLM-as-a-judge with MT-Bench and Chatbot Arena, Neurips D&B Track 2023
>
> ---
>
> > Q1. Do the authors conduct any case studies to illustrate how the model output changes from incorrect to correct? Is the improvement mainly driven by format-related instructions in the optimized system prompt, as suggested by its inclusion of output format in Appendix?
>
> This is an excellent suggestion, and based on it, we conduct additional qualitative case studies provided below, to show how MetaSPO corrects errors. Specifically, in a reasoning task (called Colored Object), the model with the default system prompt yields an incorrect answer (“red”) despite correct formatting, while the model with the system prompt from MetaSPO produces the correct answer (“brown”) by reasoning through the item order. Notably, many optimized system prompts (such as other Medical and Reasoning domain examples in Table 18) do not include format instructions, which confirms that performance gains stem not primarily from output formatting alone, but rather from improved task understanding and reasoning behavior encouraged by the optimized system prompt.
>
> **Case 1: Colored Object (Reasoning)**
> ```python
> User Prompt with question:
> Read the following questions carefully and provide the correct answers based on the details supplied in each input. Ensure your answers accurately reflect the conditions described and count or describe the objects as requested.
> <Question>
> On the floor, you see several items arranged in a row: a brown paperclip, a mauve necklace, a turquoise dog leash, a red jug, and a teal pen. What is the color of the item furthest from the pen?
> </Question>
> At the end present your answer in <answer> and </answer>.
>
> Ground Truth: Brown
>
> MetaSPO Response(Correct):
> To determine the color of the item furthest from the pen, we need to consider the arrangement of the items. Since the items are arranged in a row, we can deduce the order from left to right.
> The items in the order are:
> 1. Brown paperclip
> (...)
> 5. Teal pen
> The pen is at the end of the row, so the item furthest from the pen is the brown paperclip.
> <answer> Brown </answer>
>
> Default Response (Wrong):
>  To determine the color of the item furthest from the pen, we need to consider the arrangement of the items. Since the items are arranged in a row, we can count from the pen to the other end.
> 1. The pen is at one end.
> 2. The teal pen is closest to the pen.
> (...)
> 6. The red jug is at the farthest end.
> So, the item furthest from the pen is the red jug.
> <answer>Red</answer>
> ```
>
> **Case 2: Dental (Medical)**
> ```python
> User Prompt with question:
> For each of the following questions, select the correct answer from the provided options and indicate your choice clearly.
> <Question>
> Functional appliances in skeletal class II are indicated when:
> Options:
> A. Maxilla is normal, mandible is retrognathic
> B. Maxilla is prognathic
> C. Mandible is retrognathic in a mature individual
> D. When severe crowding is present
> </Question>
> At the end present your answer in <answer> and </answer>.
>
> Ground Truth: A
>
> MetaSPO Response(Correct):
> <answer> A </answer>
> Rationale: Functional appliances in skeletal class II are (...)
>
> Default Response(Wrong):
> <answer> C </answer>
> ```
>
> ---
>
> > Q2. Have the authors tried applying their method to more capable models (e.g., GPT-4o-latest or o3) or more challenging tasks (e.g., AIME 2024 or LiveBench)?
>
> In response to your question, we additionally conduct experiments with GPT-4o, evaluating the system prompt optimized from GPT-4o mini to GPT-4o without further optimization, and as shown in Table B.2 below, MetaSPO achieves the highest average score over baselines, indicating its strong transferability to even one of the strongest models.
>
> Also, for challenging tasks, while our evaluation already includes multiple challenging reasoning benchmarks, which require multi-step inference and compositional reasoning, we further conduct the suggested experiments on LiveBench [2], evaluating the system prompt optimized for the reasoning domain using GPT-4o mini and reporting its performance on the reasoning category of LiveBench [2]. The results in Table B.3 then show that MetaSPO outperforms baselines, highlighting its effectiveness on complex reasoning scenarios.
>
> Table B.2. Unseen generalization results of GPT-4o with system prompts optimized from GPT-4o mini
> ||Review|Analysis|||Reasoning|||
> |:---:|:---:|:---:|:---:|:---:|:---:|:---:|:---:|
> ||Electronics|Pet|Sports|Count|Epistemic|Color|Avg.|
> |Default|73.4|75.0|69.2|78.2|81.6|98.0|79.2|
> |CoT|73.0|76.0|70.6|80.0|80.0|98.4|79.7|
> |Service|70.6|75.0|66.8|82.0|81.6|98.0|79.0|
> |SPRIG|73.0|73.8|68.0|81.8|81.8|98.0|79.4|
> |MetaSPO|76.9|75.4|77.3|82.9|81.1|98.9|**82.1**|
>
> Table B.3. Result of MetaSPO on LiveBench (Nov 25, 2024) [2]
> ||LiveBench|
> |:---:|:---:|
> |Default|35.3|
> |CoT|35.8|
> |Service|36.1|
> |SPRIG|31.7|
> |MetaSPO|**39.1**|
>
> [2] LiveBench: A Challenging, Contamination-Limited LLM Benchmark, ICLR 2024
>
> ---
>
> > Q3. Could the authors provide detailed results using only ten user prompts (e.g., in the form of a table similar to Table 1) to isolate the impact of system prompt optimization?
>
> Thank you for your insightful suggestion. As suggested, we evaluate performance with only the ten user prompts without any system prompt (to isolate its effect), and report the results in the “None” row of Table B.3. From this, we observe that it shows comparable performance with the default system prompt and substantially lower performance than our optimized MetaSPO prompt. We will make sure to include this result in the revision.
>
> Table B.4: Results on the unseen generalization scenario, where None refers to using only the user prompts without any system prompt, compared against Default and MetaSPO.
> |Domain||Medical||||Review|Analysis||Reasoning||Safety||Grounding|||
> |:---:|:---:|:---:|:---:|:---:|:---:|:---:|:---:|:---:|:---:|:---:|:---:|:---:|:---:|:---:|:---:|
> |Methods|Ana.|Ped.|Den.|Sur.|Ele.|Pet|Spo.|Cou.|Epi.|Col.|A.H.|Eth.|N.Q.|Web.|Avg.|
> |Default|36.1|38.9|25.8|32.3|41.3|41.5|29.3|43.5|28.3|56.6|21.2|28.7|15.1|11.6|32.2|
> |None|40.0|36.1|31.0|33.9|38.5|39.4|27.7|42.0|34.7|59.0|19.1|27.8|13.3|10.5|32.4|
> |MetaSPO|**48.9**|**46.7**|**36.4**|**40.0**|**61.8**|**64.9**|**61.5**|**47.1**|**43.0**|**66.6**|**29.1**|**43.9**|**19.1**|**13.7**|**44.5**|
>
>
> ---
>
> > Q4. In Figure 6, why does the performance of the default system prompt vary across different iterations and example counts?
>
> We observe that the performance of the default system prompt generally improves with more iterations and higher example counts, as it increases the opportunities to adapt and incorporate relevant information from more computation. However, the performance does not increase monotonically, which we attribute to the inherent variance in LLM outputs and the stochastic nature of the prompt optimization technique (namely, ProTeGi) that may yield instability. Specifically, we found that it has an average standard deviation of 0.015 across different example counts, which reflects the variance in the results of Figure 6.

---

> ### Author Response · Authors · 2025-08-07
>
> Dear Reviewer bNQ9,
>
> Thank you once again for your thoughtful feedback, which has been instrumental in helping us improve our work. We have made every effort to carefully address all comments thoroughly, and sincerely hope you might have a chance to take a quick look before the discussion period ends. Your comments truly mean a lot and are incredibly valuable to us.
>
> Warm regards, Authors

---

> ### Author Response · Authors · 2025-08-08
>
> Dear Reviewer bNQ9,
>
> Thank you again for your valuable time and feedback on our paper. As the discussion period is ending soon, we wanted to gently remind you about our response and take this opportunity to briefly summarize how we have diligently addressed all the points you raised:
> - We pointed out that our empirical results provide concrete and sufficient support for all our claims (e.g., the efficacy of meta-learning and its generalization); thus, we believe that a theoretical analysis is neither necessary nor does its absence undermine our work.
> - We demonstrated effectiveness on subjective tasks, such as Roleplay and Writing.
> - We confirmed generalization to stronger models (GPT-4o) and more challenging benchmarks (LiveBench).
> - We provided case studies, showing that performance improvement comes from enhanced task understanding and reasoning, not merely from output formatting.
> - We added additional experiments using only the 10 user prompts to isolate the effect of the system prompt.
> - We clarified that the performance variation in Fig. 6 is due to the stochastic nature of user prompt optimization, while the overall trend is consistently upward.
>
> We sincerely hope you might take a moment to see whether our rebuttal has sufficiently addressed your concerns, or if there are any remaining points you would like us to elaborate on. We would be happy to continue the discussion should you have further comments. Otherwise, we sincerely look forward to and would greatly value your updated evaluation.
>
> Warm regards, Authors

---

### Official Review · Reviewer_Vnef · 2025-07-05

**Clarity:** 3
**Significance:** 3
**Originality:** 3
**Rating:** 4
**Confidence:** 4

**Summary:**

The paper targets the optimization of system prompt and designs a meta-learning framework to optimize system and user prompts alternatively. Extensive experiment results demonstrate the effectiveness of the proposed framework. The optimized system prompt generalizes well to unseen tasks and enables faster adaptation of user prompts.

**Questions:**

See the limitation section.

**Ethical Concerns:**

["NO or VERY MINOR ethics concerns only"]

**Final Justification:**

The issues raised in my reviews have been resolved in the rebuttal. Therefore, I vote for the acceptance of this paper.

**Limitations:**

Yes.

**Quality:**

3

**Strengths And Weaknesses:**

S1: The experiments are very comprehensive in this paper, and the results look strong.
S2: The writing is clear and easy to follow.
S3: The motivation of optimizing system prompts with awareness of user prompts is clear and intuitive.

---
W1: It would be great if the authors could provide some qualitative insights about the generalizable knowledge in the optimized system prompts, as mentioned in Line 259.
W2: Currently, the experiments are limited to using Llama-3.2-3B as the generating model and GPT-4o-mini as the prompt optimizer. I recommend that authors expand the scope by testing the framework on larger LLMs and leveraging open-sourced LLMs for prompt optimization.
W3: It would be great to break Section 4.2 into several subsections. It is currently too long.

---

> ### Author Rebuttal · Authors · 2025-07-31
>
> We thank you very much for your thoughtful comments. We are grateful that you found our motivation for system prompt optimization with awareness of user prompts is clear and intuitive, our experiments are very comprehensive with strong results, and the writing is clear. In this response, we have made every effort to address all your comments faithfully.
>
> ---
>
> > W1. It would be great if the authors could provide some qualitative insights about the generalizable knowledge in the optimized system prompts, as mentioned in Line 259.
>
> This is a very relevant proposition, and in fact, we present qualitative results about the generalizable knowledge in the optimized system prompts in Table 18. Specifically, the system prompts optimized consistently include a structured format that includes a clear role definition (e.g., you are a knowledgeable and analytical assistant designed to process and analyze information across a broad range of topics) and abstract task framing (e.g., your primary objective is to analyze and process information critically), which can be transferable across various tasks. We believe that these generalizable patterns emerge from optimization over multiple tasks via meta-learning, and the results of cross-domain and cross-model experiments in Figure 5 and Table 3 further support this. We will highlight it in the revision.
>
> ---
>
> > W2. The experiments are limited to using Llama 3.2 (3B) as the generating model and GPT-4o mini as the prompt optimizer. I recommend that authors expand the scope by testing the framework on larger LLMs and leveraging open-sourced LLMs for prompt optimization.
>
> We would like to point out that, in addition to Llama 3.2 (3B), we also present experiment results with the stronger generating models in Table 3, including Llama (8B), Qwen 2.5 (7B), and GPT-4o mini, from which we observe that MetaSPO consistently outperforms baselines, confirming its generalizability and robustness to various models. However, to more fully address your suggestion, we additionally conduct experiments with GPT-4o, evaluating the system prompt optimized from GPT-4o mini to GPT-4o without further optimization, and as shown in Table A.1 below, MetaSPO achieves the highest average score over baselines, indicating its strong transferability to even the strongest model. Lastly, we would like to refer to the results in Table 15, showing that MetaSPO is compatible with open-source models, such as Llama (8B) and Llama (70B), when they were used as prompt optimizers.
>
> Table A.1. Unseen generalization results of GPT-4o with system prompts optimized from GPT-4o mini
> | 　 | Review | Analysis |  |  | Reasoning |  |  |
> |:---:|:---:|:---:|:---:|:---:|:---:|:---:|:---:|
> | 　 | Electronics | Pet | Sports | Count | Epistemic | Color | Avg. |
> | Default | 73.4 | 75.0 | 69.2 | 78.2 | 81.6 | 98.0 | 79.2 |
> | CoT | 73.0 | 76.0 | 70.6 | 80.0 | 80.0 | 98.4 | 79.7 |
> | Service | 70.6 | 75.0 | 66.8 | 82.0 | 81.6 | 98.0 | 79.0 |
> | SPRIG | 73.0 | 73.8 | 68.0 | 81.8 | 81.8 | 98.0 | 79.4 |
> | MetaSPO | 76.9 | 75.4 | 77.3 | 82.9 | 81.1 | 98.9 | **82.1** |
>
> ---
>
> > W3. It would be great to break Section 4.2 into several subsections. It is currently too long.
>
> Thank you for your valuable suggestion, and we apologize for the inconvenience. We will split it into multiple subsections (for example, Unseen Generalization, Test-Time Adaptation, and Additional Analysis) to improve the quality and clarity of the presentation in the revision.

---

> > ### Comment · Reviewer_Vnef · 2025-08-04
> >
> > Thanks for your response. I would keep the rating and vote for acceptance of the paper.

---

> > > ### Author Response · Authors · 2025-08-04
> > >
> > > Thank you once again for your thoughtful review and feedback, and for taking the time to review our responses. We truly appreciate your support in voting for the acceptance of our paper, and we will ensure that all of your suggestions are incorporated into the final revision. Thank you again!

---

### Author Response · Authors · 2025-08-06

Dear Reviewers,

We greatly appreciate your time and efforts in reviewing our paper, as well as your insightful and constructive comments. We would like to inform you that we have diligently responded to each of your comments in great detail.

We fully understand that your time is extremely valuable and that you are likely managing many responsibilities. However, as the discussion window will be closed in the near future, we would be deeply grateful if you could kindly take a moment to review our response. Please let us know if you have any additional points, which would be immensely valuable to us.

Thank you once again for your service, and we look forward to your response!


Warm regards, Authors

---

### Note · Authors · 2025-08-12

Dear All,

We wanted to sincerely thank all reviewers for the time, effort, and constructive feedback, as well as the ACs, SACs, and PCs for their consideration during the decision process.

We wanted to recap our contribution: we introduce bilevel system prompt optimization, a novel problem setting that (unlike prior work focused on task-specific user prompts) targets the optimization of task-agnostic system prompts via meta-learning, which enhances robustness and transferability across diverse user prompts and unseen tasks. We are truly grateful that all reviewers recognized these contributions, highlighting the clarity, importance, and soundness of our objective for system prompt optimization (Reviewers Vnef, bNQ9, rqST, tfVR), the comprehensiveness of our experiments (Vnef, bNQ9), the demonstrated effectiveness on unseen tasks (rqST, tfVR), the breadth and depth of our analyses (bNQ9, rqST), the benefits in generalization and faster optimization (rqST), and the clarity of our writing (Vnef, rqST).

In the meantime, we have also made every effort to address all concerns raised, to ensure that each point was thoroughly responded to with additional experiments and clarifications. Specifically, as suggested, we extended our evaluation to more capable models (e.g., GPT-4o), demonstrated further applicability to both subjective (MT-Bench) and challenging (LiveBench) tasks, conducted a detailed cost analysis confirming the efficiency, and showed that MetaSPO complements and enhances existing user prompt optimization methods.

Following our initial response, we are grateful that two reviewers (Vnef and rqST) voted for acceptance without raising any concerns. In addition, we also reached out to the other two reviewers to invite any remaining questions or clarifications; however, one (tfVR) noted they would consider our answers when revising their evaluation, but did not raise further points, and the other (bNQ9) did not respond. In light of this absence of any new concerns, we are very confident that all points raised have been fully and constructively addressed.

We firmly believe that our contribution to system prompt optimization via meta-learning offers substantial and long-lasting value to the community. Thus, we respectfully ask that all the aforementioned points be taken into account, and in the meantime, we will make sure to incorporate all the suggestions into the final version.

Thank you again for your time and consideration.

Warm regards, Authors

---

### Decision · Program_Chairs · 2025-09-17

**Decision:**

Accept (poster)

**Comment:**

This paper proposes MetaSPO, a bilevel prompt optimization framework that jointly optimizes system prompts and task-specific user prompts for large language models. By alternating between updating user prompts based on task-specific failures and updating the system prompt using aggregated failure patterns, the method improves performance on unseen tasks and accelerates adaptation of user prompts. The approach is evaluated across 14 datasets spanning five task domains, demonstrating consistent gains over baseline prompting methods. The paper is well-written, clearly motivated, and the experiments are comprehensive.

Reviewers generally agree that the paper is technically solid and empirically strong, with intuitive motivation and clear benefits of system prompt optimization. Strengths include the comprehensive evaluation, clarity of presentation, and demonstrated generalization. Weaknesses mostly concern novelty, theoretical grounding, and scope: the method resembles a search-based optimization rather than classical meta-learning, relies on strong external analyzers, and its effectiveness on subjective or open-ended tasks is untested. Minor concerns were raised regarding implementation details and the computational cost relative to task-only optimization.

Despite these limitations, the authors’ rebuttal clarified most concerns, particularly regarding experimental validity and implementation choices. The practical significance of jointly optimizing system and user prompts is clear, showing improved efficiency and performance across tasks. On balance, the strengths and impact of the work outweigh the concerns, and I recommend acceptance.